# YTHDC1 mediates nuclear export of N$^6$-methyladenosine methylated mRNAs

Ian A Roundtree[1,2,3,4], Guan-Zheng Luo[1,2,3], Zijie Zhang[1,2,3], Xiao Wang[1,2,3], Tao Zhou[5], Yiquang Cui[6], Jiahao Sha[6], Xingxu Huang[5], Laura Guerrero[1,2,3], Phil Xie[1,2,3], Emily He[1,2,3], Bin Shen[6]*, Chuan He[1,2,3]*

[1]Department of Chemistry, University of Chicago, Chicago, United States; [2]Department of Biochemistry and Molecular Biology, and Institute for Biophysical Dynamics, University of Chicago, Chicago, United States; [3]Howard Hughes Medical Institute, University of Chicago, Chicago, United States; [4]University of Chicago Medical Scientist Training Program, Chicago, United States; [5]School of Life Science and Technology, ShanghaiTech University, Shanghai, China; [6]Department of Histology and Embryology, State Key Laboratory of Reproductive Medicine, Nanjing Medical University, Nanjing, China

**Abstract** *N$^6$*-methyladenosine (m$^6$A) is the most abundant internal modification of eukaryotic messenger RNA (mRNA) and plays critical roles in RNA biology. The function of this modification is mediated by m$^6$A-selective 'reader' proteins of the YTH family, which incorporate m$^6$A-modified mRNAs into pathways of RNA metabolism. Here, we show that the m$^6$A-binding protein YTHDC1 mediates export of methylated mRNA from the nucleus to the cytoplasm in HeLa cells. Knockdown of YTHDC1 results in an extended residence time for nuclear m$^6$A-containing mRNA, with an accumulation of transcripts in the nucleus and accompanying depletion within the cytoplasm. YTHDC1 interacts with the splicing factor and nuclear export adaptor protein SRSF3, and facilitates RNA binding to both SRSF3 and NXF1. This role for YTHDC1 expands the potential utility of chemical modification of mRNA, and supports an emerging paradigm of m$^6$A as a distinct biochemical entity for selective processing and metabolism of mammalian mRNAs.
DOI: https://doi.org/10.7554/eLife.31311.001

*For correspondence:
binshen@njmu.edu.cn (BS);
chuanhe@uchicago.edu (CH)

Competing interests: The authors declare that no competing interests exist.

## Introduction

*N$^6$*-methyladenosine, the most common internal modification of eukaryotic mRNA, is associated with the maturation, translation, and eventual decay of protein-coding transcripts (*Roundtree et al., 2017*; *Tuck, 1992*). 'Reader' proteins of the YTH and HNRNP families mediate many of the properties of m$^6$A-methylated transcripts through methyl-specific RNA binding. Within the cytoplasm, YTHDF1 and YTHDF2 facilitate m$^6$A-dependent translation initiation and m$^6$A-dependent mRNA decay, respectively, under normal and stress conditions (*Batista et al., 2014*; *Du et al., 2016*; *Geula et al., 2015*; *Liu et al., 2014*; *Wang et al., 2014a, 2014b, 2015*; *Zhou et al., 2015*). Additional roles in m$^6$A-based regulation of cap-independent translation have been described for eIF3 as well as for the catalytic methyltransferase component METTL3, each functioning to promote the translation of methylated mRNAs (*Lin et al., 2016*; *Meyer et al., 2015*). The functions of these 'reader' proteins provide a step-wise, mechanistic understanding of the characteristic relationship between m$^6$A modification and mRNA translation and decay. Within the nucleus, YTHDC1, HNRNPA2B1 and HNRNPC bind to m$^6$A-modified RNAs and facilitate splice site selection (*Alarcón et al., 2015*; *Liu et al., 2015*; *Xiao et al., 2016*; *Xu et al., 2014*), individual examples of which are required for vertebrate development (*Haussmann et al., 2016*; *Lence et al., 2016*).

However, the relationship between m$^6$A methylation and other nuclear events has yet to be investigated fully. Specifically, the direct link between nuclear mRNA processing and the pool of cytoplasmic transcripts available for translation in the cytoplasm has remained elusive.

The physical segregation of biological processes into nuclear and cytoplasmic compartments is a fundamental feature of eukaryotic organisms, and necessitates efficient systems for shuttling mRNA to sites of active translation. The functional coupling of the transcription, 5' capping, splicing, 3' polyadenylation, and nuclear export of mRNA ensures that genetic information is properly passed from DNA to protein (*Moore and Proudfoot, 2009*), as exemplified by the scaffolding functions of RNA polymerase II throughout transcription (*Egloff and Murphy, 2008*). The existence of multifunctional adaptor proteins that have roles in the pre-mRNA splicing and export of mature mRNA suggests that mRNA nuclear processing and transport are regulated by the dynamics of protein–RNA interactions in eukaryotes (*Huang and Steitz, 2001*; *Huang et al., 2003*; *Müller-McNicoll et al., 2016*; *Rodrigues et al., 2001*; *Strässer and Hurt, 2001*; *Valencia et al., 2008*). It has become clear that the regulation of mRNA export occurs in response to various stimuli, and is affected by both *cis* elements within mRNA sequences and the specificity of *trans* factors such as RNA-binding proteins (*Carmody et al., 2009*; *Wickramasinghe and Laskey, 2015*).

Within the nucleus, m$^6$A modification of mRNA has emerged as a previously unrecognized regulator of RNA structure (*Liu et al., 2015*; *Spitale et al., 2015*) and of protein–RNA interactions. m$^6$A has long been considered to play critical roles in constitutive splicing as well as in processing-linked transport of cellular and viral mRNAs (*Finkel and Groner, 1983*; *Fustin et al., 2013*; *Lichinchi et al., 2016*; *Stoltzfus and Dane, 1982*), suggesting fundamental roles for mRNA methylation within the nucleus in addition to those in pre-mRNA splicing. Treatment with the S-adenosylhomocysteine analog S-tubercidinylhomocysteine (STH) reduces m$^6$A levels by over 80% in HeLa cells and results in delayed nuclear export of mature mRNA (*Camper et al., 1984*), whereas depletion of the m$^6$A demethylase ALKBH5 results in rapid cytoplasmic appearance of mRNA (*Zheng et al., 2013*), further suggesting that m$^6$A is directly involved in nuclear-to-cytoplasmic localization. Despite evidence for m$^6$A-dependent nuclear transport, the underlying mechanism and pathway have never been revealed.

In this work, we demonstrate that the nuclear m$^6$A 'reader' protein YTHDC1 mediates the export of methylated mRNAs. Knockdown of YTHDC1 results in nuclear accumulation of methylated mRNAs, accompanied by cytoplasmic depletion of these mRNAs in HeLa cells, whereas binding by YTHDC1 promotes the redistribution of transcripts to the cytoplasm. YTHDC1 incorporates target mRNAs into the nuclear export pathway by interaction with SRSF3, delivering RNA to the splicing and adaptor protein. Association of YTHDC1 targets with SRSF3 is required for YTHDC1-dependent export. By performing analysis of subcellular mRNA populations, we show that YTHDC1 utilizes known interactions with SRSF3 to selectively mediate the entry of methylated mRNAs into known routes for export to the cytoplasm.

## Results

m$^6$A promotes the clearance of cytoplasmic and nuclear mRNAs and is known to facilitate rapid clearance of mRNAs from the cytoplasm due to direct binding of YTHDF2 and recruitment of the CCR4–NOT deadenylase complex (*Du et al., 2016*; *Wang et al., 2014a*). We confirmed this cytoplasmic role by treating HeLa cells with Actinomycin D and then monitoring m$^6$A methylation in cytoplasmic mRNA (polyA selected, ribosomal RNA-depleted transcripts) by liquid chromatography coupled to tandem mass spectrometry (LC-MS/MS) (*Figure 1—figure supplement 1*). Cells treated with control short interfering RNA (siRNA) preferentially cleared m$^6$A-containing mRNAs, as indicated by a reduction in the m$^6$A/A ratio in the absence of new transcription (*Figure 1A*, black). These data confirm that in the absence of newly transcribed RNA, methylated mRNAs are selectively cleared from the cytoplasmic pool, allowing us to calculate an apparent half-life of mRNA methylation from the decreasing m$^6$A/A signal. This ensemble measurement allows us to identify changes in the relative rate at which methylated transcripts are cleared under two conditions without determining the absolute half-lives of mRNAs. Knockdown of the cytoplasmic protein YTHDF2 resulted in increased m$^6$A levels (t = 0), and extended the apparent half-life for m$^6$A in cytoplasmic mRNA by ~4 fold (*Figure 1A*, gray, *Figure 1—figure supplement 1A*), as would be expected in the absence of the m$^6$A-specific mRNA decay factor. We next asked whether m$^6$A methylation similarly

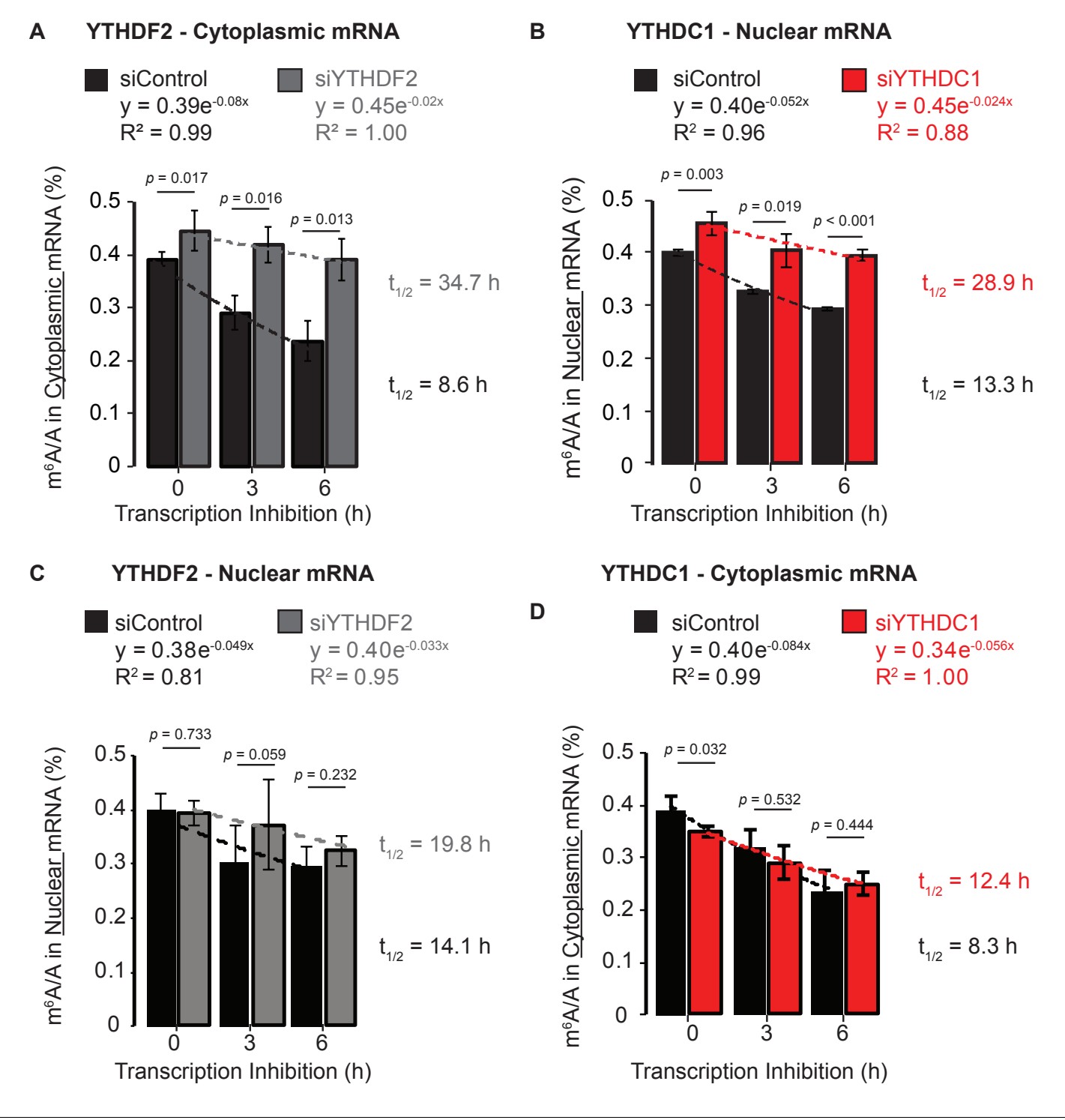

**Figure 1.** Methylated mRNAs are selectively cleared from the cytoplasm by YTHDF2 and from the nucleus by YTHDC1. (A) m6A methylation in cytoplasmic mRNA analyzed at 0, 3, and 6 hr after transcription inhibition. Error bars represent mean ± standard deviation, n = 4, two-sided *t*-test with equal variance. Curves fit to exponential decay. (B) m6A methylation in nuclear mRNA analyzed at 0, 3, and 6 hr after transcription inhibition. Error bars represent mean ± standard deviation, n = 4, two-sided *t*-test with equal variance. Curves fit to exponential decay. (C) m6A methylation in nuclear mRNA analyzed at 0, 3, and 6 hr after transcription inhibition. Error bars represent mean ± standard deviation, n = 4, two-sided *t*-test with equal variance. Curves fit to exponential decay. (D) m6A methylation in cytoplasmic mRNA analyzed at 0, 3, and 6 hr after transcription inhibition. Error bars represent mean ± standard deviation, n = 4, two-sided *t*-test with equal variance. Curves fit to exponential decay.

*Figure 1 continued on next page*

*Figure 1 continued*

DOI: https://doi.org/10.7554/eLife.31311.002

The following source data and figure supplements are available for figure 1:

**Source data 1.** Methylated mRNAs are selectively cleared from the cytoplasm by YTHDF2 and from the nucleus by YTHDC1.

DOI: https://doi.org/10.7554/eLife.31311.005

**Figure supplement 1.** Representative western blots for siRNA knockdown efficiency and northern blot nuclear and cytoplasmic RNA isolation.

DOI: https://doi.org/10.7554/eLife.31311.003

**Figure supplement 1—source data 1.** Calibration of LC-MS/MS.

DOI: https://doi.org/10.7554/eLife.31311.004

facilitates the expedited clearance of mRNA from cell nuclei. Following inhibition of transcription, nuclear m$^6$A levels in mRNA decreased over a six-hour time course, suggesting that selective processing and clearance of these mRNAs is common to both the nuclear and the cytoplasmic compartments (*Figure 1B*, black). Knockdown of the only nuclear member of the YTH family and known m$^6$A-binding protein, YTHDC1, resulted in increased m$^6$A levels in nuclear mRNA (t = 0) and in a decreased rate of clearance of m$^6$A methylation in mRNA, extending the apparent half-life of methylation more than 2-fold (*Figure 1B*, red, *Figure 1—figure supplement 1B and C*). Analysis of mRNA from the nucleus showed no difference between control and YTHDF2 knockdown samples, confirming that the dominant role of YTHDF2 takes place in the cytoplasm under normal growth conditions (*Figure 1C*). Knockdown of YTHDC1 resulted in decreased cytoplasmic levels of m$^6$A, but these levels approached control levels over the time course (*Figure 1D*). These data suggest that while YTHDF2 mediates cytoplasmic decay of methylated mRNA transcripts, YTHDC1 mediates nuclear clearance of mRNAs and affects their resulting cytoplasmic abundance.

## YTHDC1 mediates nuclear to cytoplasmic transport of methylated mRNAs

We further examined the relationship between YTHDC1 and the subcellular localization of methylated mRNAs by LC-MS/MS. Knockdown of YTHDC1 does not affect the m$^6$A/A ratio in mRNA transcripts from whole cell lysate, but results in accumulation of m$^6$A in nuclear mRNA and in an accompanying cytoplasmic depletion of m$^6$A levels (*Figure 2A*). These effects occurred without changes in the expression of YTHDF2, or of key components of the nuclear export machinery, the mRNA export receptor NXF1 or the adaptor protein ALYREF (*Figure 2B*). We can directly observe the effect of YTHDC1 knockdown on mRNA distribution by fluorescence *in situ* hybridization of polyA RNAs in HeLa cells, in which YTHDC1 knockdown results in increased signal intensity in the nuclear region and depletion in the cytoplasmic regions (*Figure 2C and D*), suggesting a positive role for YTHDC1 in nuclear mRNA export.

Overexpression of YTHDC1 reduced levels of m$^6$A in nuclear mRNA, suggesting that it contributes directly to the clearance of these transcripts, but did not result in increased levels of cytoplasmic m$^6$A as anticipated (*Figure 2E*, *Figure 2—figure supplement 1A*). One possibility is that YTHDC1 may interact with components of the nuclear RNA decay machinery to mediate nuclear mRNA decay. Indeed, YTHDC1 has been reported to interact with ZCCHC8, a component of the NEXT complex (*Lubas et al., 2011*), an interaction that we verified by western Blot. We found a robust interaction between YTHDC1 and ZCCHC8, but no interaction between YTHDC1 and MTR4 (*Figure 2—figure supplement 1B*). Knockdown of these components did not produce an increase in nuclear RNA that is depleted for ribosomal transcripts (*Figure 2—figure supplement 1C and D*). These results suggest that the accumulation of mature mRNA transcripts with YTHDC1 knockdown may not be attributed to nuclear decay. We then overexpressed YTHDC1 following knockdown of YTHDF2 (*Figure 2—figure supplement 1E*). Under these conditions, we observed nuclear depletion as expected, as well as an increase in cytoplasmic m$^6$A levels compared to empty vector control, suggesting that YTHDC1 mediates nuclear export of methylated mRNAs (*Figure 2F*) and that this nuclear localized protein can affect subcellular distribution of methylated transcripts (*Figure 2—figure supplement 1F*). We compared the bulk transcriptional activity of HeLa cells treated with siRNA against YTHDC1 with control and YTHDC1-overexpressing lines, and found that these perturbations of YTHDC1 did not dramatically affect the incorporation of 4-thio-urudine in a 1-hr

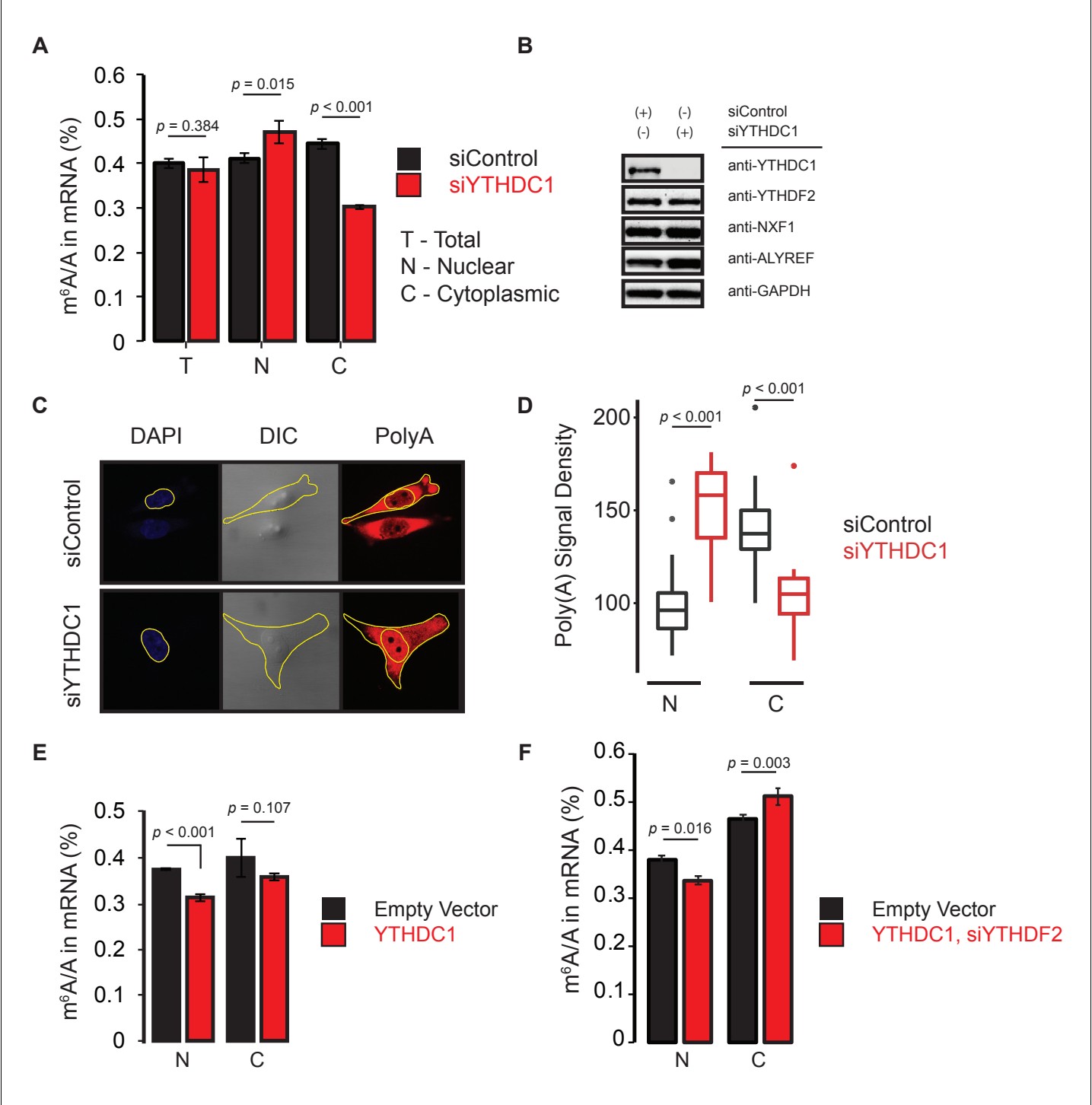

**Figure 2.** YTHDC1 mediates subcellular localization of methylated mRNAs. (**A**) Quantification of m$^6$A methylation in total (T), nuclear (N), and cytoplasmic (C) mRNA. Error bars represent mean ± standard deviation, n = 4, two-sided *t*-test with equal variance. (**B**) Representative western blot of YTHDF2 and members of the mRNA export pathway following knockdown of YTHDC1. (**C**) Representative analysis of polyA imaging. Nuclei were defined using DAPI signal. Cytoplasmic regions were defined by subtracting nuclear signal from total signal, as defined by DIC imaging. (**D**) Quantification of polyA signal density. n = 25, two-sided *t*-test. (**E,F**) Quantification of m$^6$A methylation in nuclear and cytoplasmic mRNA. Error bars represent mean ± standard deviation, n = 4, two-sided *t*-test with equal variance.

DOI: https://doi.org/10.7554/eLife.31311.006

The following source data and figure supplement are available for figure 2:

**Source data 1.** YTHDC1 mediates subcellular localization of methylated mRNAs.

*Figure 2 continued on next page*

*Figure 2 continued*

DOI: https://doi.org/10.7554/eLife.31311.008

**Figure supplement 1.** YTHDC1 does not contribute to nuclear mRNA decay or transcription.

DOI: https://doi.org/10.7554/eLife.31311.007

pulse period (*Figure 2—figure supplement 1G and H*), suggesting that YTHDC1 functions post-transcriptionally.

## YTHDC1 affects subcellular abundance of target transcripts

In order to appreciate the function of YTHDC1 on the subcellular distribution of its target transcripts, we employed RIP-seq as a complimentary method of identifying high-confidence RNA substrates when combined with our previously reported PAR-CLIP data (*Peritz et al., 2006*; *Xu et al., 2014*). We generated two YTHDC1 RIP-seq data sets; one with RNA purified by polyA selection, the other with ribosomal depletion. The two datasets show good correlation overall, with major deviation a result of enrichment for methylated transcripts lacking polyA tails in the ribosomal RNA-depleted sample (*Figure 3—figure supplement 1A*). We identified targets as transcripts that were bound in each previous replicate of PAR-CLIP and enriched at least two-fold in each replicate of RIP-seq. These criteria produced over 700 targets of YTHDC1 (*Figure 3—figure supplement 1B*). We then analyzed the subcellular transcriptome of HeLa cells following knockdown of YTHDC1. Consistent with our mass spectrometry data, knockdown of YTHDC1 does not affect the abundance of target transcripts relative to non-target mRNAs in whole-cell samples (*Figure 3A*). Notably, target abundance is unchanged from control levels in YTHDC1 knockdown cells (*Figure 3—figure supplement 1C*). In the nuclear RNA sample, however, target abundance is increased following knockdown of YTHDC1, as indicated by a right shift in the cumulative fraction of these transcripts compared to non-target mRNAs (*Figure 3B*). An accompanying decrease in cytoplasmic target abundance suggests that YTHDC1 mediates the subcellular distribution of its target mRNAs by promoting nuclear to cytoplasmic transport of bound transcripts (*Figure 3C*). This shift in subcellular distribution is also observable for PAR-CLIP and RIP-seq targets of YTHDC1 (*Figure 3—figure supplement 2A–D*).

## YTHDC1 mediates nuclear export of nascent mRNA

In order to investigate the effect of YTHDC1 on the localization of newly synthesized mRNA, we utilized nascent RNA labeling using 5-ethynyl uridine. Following a 4-hr pulse with the nucleoside analog, we isolated nuclear and cytoplasmic RNA from cells treated with a control siRNA and with siRNA against YTHDC1 (*Figure 4—figure supplement 1A*). We then analyzed the subcellular abundance of several targets of YTHDC1 under control and knockdown conditions in these nascent RNA fractions (*Figure 4—figure supplement 1B*). Upon knockdown of YTHDC1, we observed nuclear accumulation of nascent mRNA for both APC and MCL1 transcripts compared to control, whereas the difference in nuclear SOX12 mRNA abundance was modest and not significant in our assay (*Figure 4A*). Cytoplasmic RNA invariably showed reduced abundance of newly synthesized YTHDC1 targets upon knockdown, suggesting that these transcripts fail to accumulate within the cytoplasm when compared to control levels (*Figure 4B*). Taken together, these data indicate that YTHDC1 is required for proper transport of mature mRNA species from the nucleus to the cytoplasm, as we utilized primer sequences specific for exon-exon junctions of mature transcripts. Knockdown of YTHDC1 did not have a significant effect on the distribution of non-target transcripts (*Figure 4—figure supplement 1C*).

## YTHDC1 affects subcellular mRNA localization independent of pre-mRNA splicing

Both $m^6A$ and YTHDC1 have been shown to affect alternative splicing patterns in mRNA (*Alarcón et al., 2015*; *Dominissini et al., 2012*; *Liu et al., 2015*; *Xiao et al., 2016*). In order to identify whether changes in subcellular localization were due to changes in alternative splicing, we performed paired-end sequencing of total RNA following knockdown of *YTHDC1*. Analysis using Mixture of Isoforms (MISO) software (*Katz et al., 2010*) identified 426 alternative splicing events with a difference in percent spliced in (psi, $\Psi$) value of 0.2 or greater (*Figure 4—figure supplement*

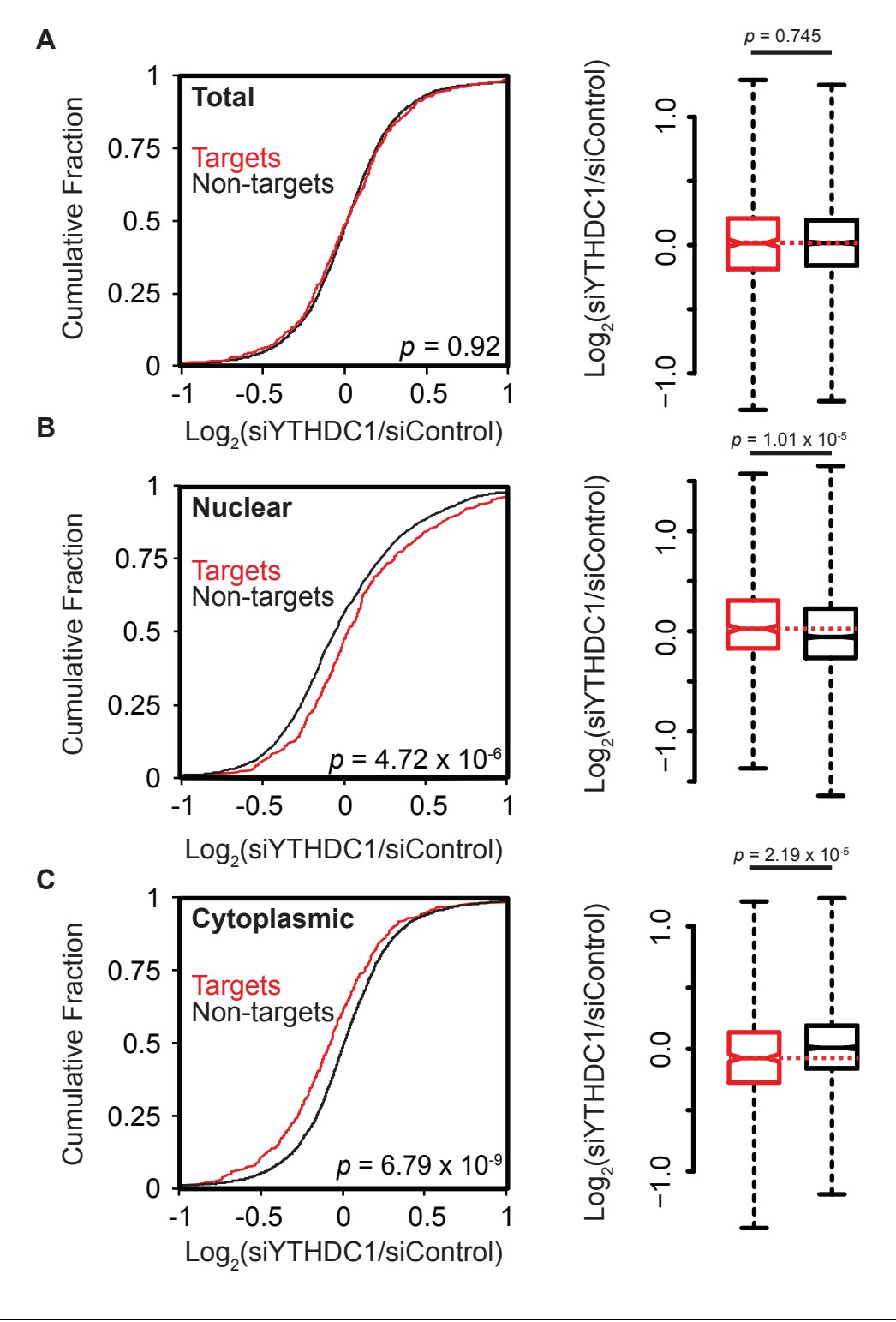

**Figure 3.** Knockdown of YTHDC1 affects the subcellular distribution of target mRNAs. (**A**) Left: Cumulative distribution of $\log_2$ fold changes in RNA expression following knockdown of *YTHDC1* in total mRNA. *P*-value calculated using the Mann-Whitney-Wilcoxon Test. Right: Boxplot representation of RNA-seq fold changes. Whiskers represent three times the interquartile range. *P*-value calculated using Welch's *T*-test. (**B**) Left: Cumulative distribution of $\log_2$ fold changes in RNA expression following knockdown of *YTHDC1* in nuclear mRNA. *P*-value calculated using the Mann-Whitney-Wilcoxon Test. Right: Boxplot representation of RNA-seq fold

*Figure 3 continued on next page*

*Figure 3 continued*

changes. Whiskers represent three times the interquartile range. *P*-value calculated using Welch's *T*-test. (**C**) Left: Cumulative distribution of $\log_2$ fold changes in RNA expression following knockdown of *YTHDC1* in cytoplasmic mRNA. *P*-value calculated using the Mann-Whitney-Wilcoxon Test. Right: Boxplot representation of RNA-seq fold changes. Whiskers represent three times the interquartile range. *P*-value calculated using Welch's *T*-test. Data represent biological replicates using unique siRNAs against *YTHDC1*.

DOI: https://doi.org/10.7554/eLife.31311.009

The following source data and figure supplements are available for figure 3:

**Source data 1.** Knockdown of YTHDC1 affects the subcellular distribution of target mRNAs.
DOI: https://doi.org/10.7554/eLife.31311.013
**Figure supplement 1.** Identification of YTHDC1 mRNA targets; target expression is unchanged following knockdown of YTHDC1.
DOI: https://doi.org/10.7554/eLife.31311.010
**Figure supplement 2.** Subcellular abundances of YTHDC1 PAR-CLIP and RIP-seq targets are affected by YTHDC1.
DOI: https://doi.org/10.7554/eLife.31311.011
**Figure supplement 2—source data 1.** Subcellular abundances of YTHDC1 PAR-CLIP and RIP-seq targets are affected by YTHDC1.
DOI: https://doi.org/10.7554/eLife.31311.012

---

*2A*, Materials and methods). However, the alternative splicing events observed following knockdown of *YTHDC1* show only weak negative correlations to those observed following knockdown of *METTL3* (*Figure 4—figure supplement 2B*) (*Alarcón et al., 2015*). Of these 426 alternative splicing events, 85 occur in our high confidence YTHDC1 targets (85/737, 11.5%, *Figure 4—figure supplement 2C*). 108 of the 426 alternative splicing events occur in YTHDC1 non-targets (108/4,848, 2.2%, *Figure 4—figure supplement 2D*). In addition, individual transcripts of *APC* and *MCL1* do not show significant changes in alternative exon usage following knockdown of *YTHDC1* (*Figure 4—figure supplement 2E*), whereas *SOX12* mRNA has only one isoform and is therefore not a candidate for regulation by potential alternative splicing events. These results show that YTHDC1 does affect alternative splicing patterns, but may play limited roles in alternative splice site selection in HeLa cells (*Ke et al., 2017*).

To confirm a role in mRNA export independent of pre-mRNA splicing, we conducted a reporter assay to study the effect of direct binding by YTHDC1. Firefly luciferase was fused to five copies of the BoxB sequence and placed under control of a Tet-off promoter (*Wang et al., 2014a*). Effector constructs derived from *YTHDC1* were fused to the lambda peptide, providing a high affinity interaction between protein and RNA. This system offers several advantages in studying the relationship between RNA binding and mRNA fate. The system is inducible and thus offers time-resolved information concerning both mRNA localization and protein production. The transcript is splicing-independent, and the RNA binding of the effector constructs is methylation-independent due to the lambda peptide-BoxB interaction. Last, the system is internally normalized by the presence of constitutively active Renilla luciferase on the same plasmid as the inducible Firefly luciferase (*Figure 4—figure supplement 3A*). Using this construct, we fused full length *YTHDC1* to the lambda peptide and monitored translation of the Firefly luciferase following a 2-hr pulse of transcription. When bound by YTHDC1 rather than by a G-S-S control effector, we observed an increase in Firefly/Renilla luciferase signal (*Figure 4C*). This effect occurs independent of $m^6A$ binding and it is maintained for the W377A mutant construct (*Figure 4—figure supplement 3D*). Analysis of mRNA distribution 4 hr after the induction of transcription showed that binding by YTHDC1 favors mRNA distribution to the cytoplasm: we observed decreased mRNA levels of Firefly/Renilla mRNA in the nucleus, and increased ratios in the cytoplasm (*Figure 4D*). We further analyzed this system using both N- and C-terminal constructs of YTHDC1, both of which feature several distinct regions of predicted low complexity (*Letunic et al., 2015*). Tethering of the C-terminal effector also produced increased translation of the Firefly reporter construct (*Figure 4E*), and biased the mRNA distribution of the Firefly transcript towards the cytoplasm (*Figure 4F*), despite both the full-length and C-terminal constructs being strictly localized to the nucleus (*Figure 4—figure supplement 3C*). Tethering by the N-terminal construct produced no significant change in translation, although Firefly mRNA is retained in the nucleus upon binding by this construct (*Figure 4—figure supplement 3E and F*).

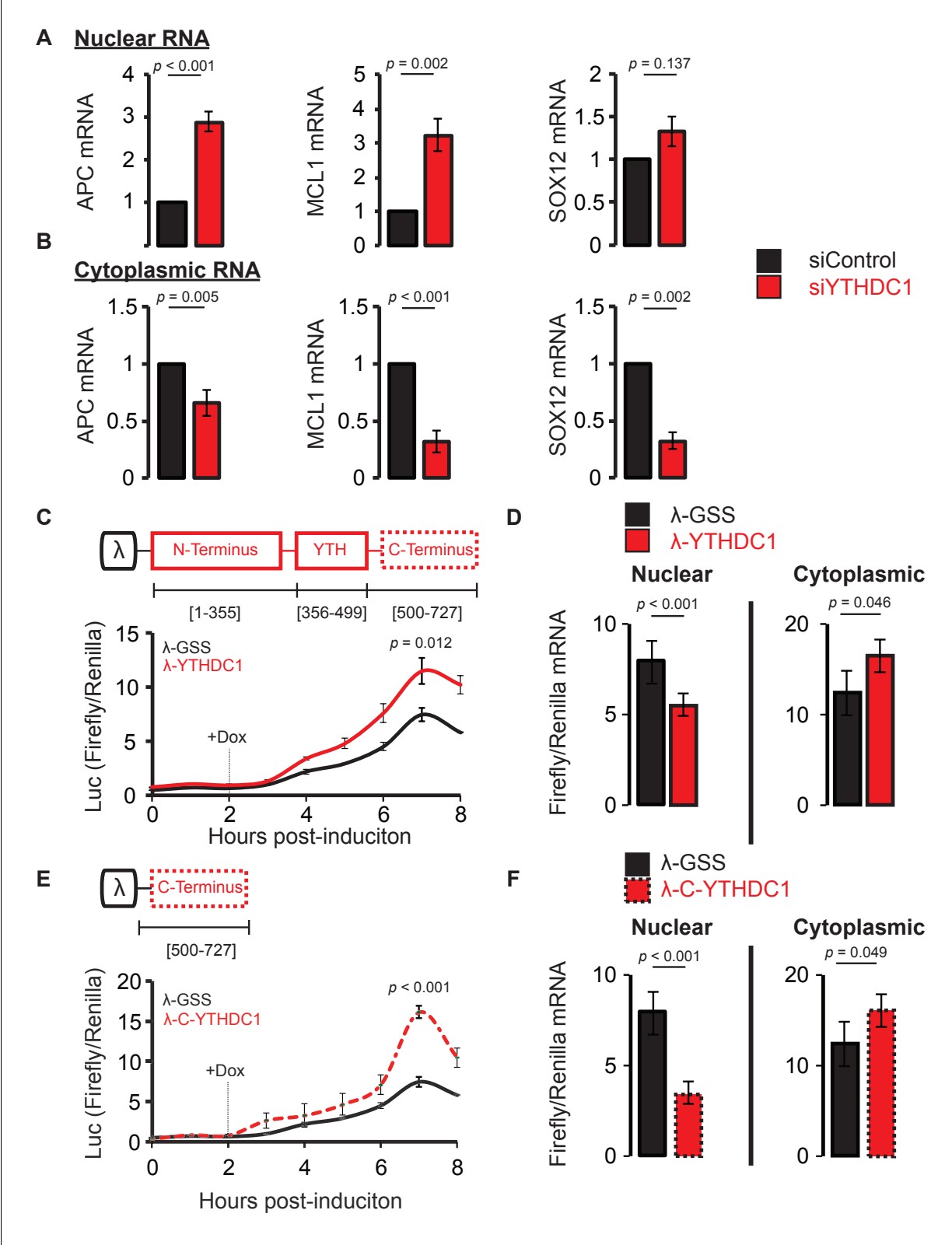

**Figure 4.** YTHDC1 mediates nuclear to cytoplasmic transport of nascent mRNA. (**A,B**) RT-qPCR of nascent mRNA in the nucleus (A) and cytoplasm (B). Error bars represent mean ± standard deviation, n = 6 (three biological replicates x two technical replicates), two-sided *t*-test with equal variance. (**C**) Translation of reporter mRNA in response to binding by the full-length YTHDC1. Error bars represent mean ± standard deviation, n = 4. (**D**) Nuclear and cytoplasmic RT-qPCR of reporter mRNA 4 hr after induction of reporter transcript. Error bars represent mean ± standard deviation, n = 4, two-sided *t*-

*Figure 4 continued on next page*

*Figure 4 continued*

test with equal variance. (E) Translation of reporter mRNA in response to binding by the YTHDC1 C-terminus. Error bars represent mean ± standard deviation, n = 4, two-sided *t*-test with equal variance. (F) Nuclear and cytoplasmic RT-qPCR of reporter mRNA 4 hr after induction of reporter transcript. Error bars represent mean ± standard deviation, n = 4, two-sided *t*-test with equal variance.

DOI: https://doi.org/10.7554/eLife.31311.014

The following source data and figure supplements are available for figure 4:

**Source data 1.** YTHDC1 mediates nuclear to cytoplasmic transport of nascent mRNA.

DOI: https://doi.org/10.7554/eLife.31311.018

**Figure supplement 1.** Nascent mRNA localization by metabolic labeling with 5-ethynyl-uridine (EU).

DOI: https://doi.org/10.7554/eLife.31311.015

**Figure supplement 2.** YTHDC1 affects alternative pre-mRNA splicing independent of roles in mRNA binding and export.

DOI: https://doi.org/10.7554/eLife.31311.016

**Figure supplement 3.** YTHDC1 reporter system.

DOI: https://doi.org/10.7554/eLife.31311.017

This result is consistent with roles for the N-terminus of YTHDC1 in pre-mRNA splicing and function of the XIST non-coding RNA (*Patil et al., 2016*; *Xiao et al., 2016*). The effects of binding by the C-terminal region of YTHDC1 represent a new role for YTHDC1 in mRNA export.

## YTHDC1 bridges m$^6$A selectivity to mRNA export via SRSF3

YTHDC1 is known to interact with members of the SR protein family, which play diverse roles in pre-mRNA splicing and export (*Huang and Steitz, 2001*; *Huang et al., 2003*; *Müller-McNicoll et al., 2016*; *Xiao et al., 2016*). We confirmed the interactions between flag-tagged YTHDC1 and several members of the SR family which have been previously reported (*Xiao et al., 2016*) (*Figure 5—figure supplement 1A*). We further confirmed the interaction between endogenous YTHDC1 and SRSF3 by immunoprecipitation (IP) of YTHDC1. Although YTHDC1 interacts with SRSF3, we did not observe any direct interaction between YTHDC1 and the export receptor NXF1, nor did we observe interaction between YTHDC1 and the common mRNA adaptor protein ALYREF (*Figure 5A*). We hypothesized that the interaction between YTHDC1 and the mRNA export pathway is mediated by export adaptor proteins. Indeed, SRSF3 is known to interact with both the splicing and the export machinery, and is known to do so in a phosphorylation-dependent manner, with hypo-phosphorylated protein interacting with the export receptor (*Huang et al., 2004*). When probing for serine phosphorylation in YTHDC1 IP samples, we did not detect phosphorylation at the molecular weight of SRSF3, indicating interaction between YTHDC1 and an export-competent form of SRSF3 that lacks serine phosphorylation. We validated the interaction between SRSF3 and NXF1 by IP of the endogenous SRSF3 and NXF1 proteins. SRSF3 robustly interacts with YTHDC1 in the presence and absence of RNA, whereas NXF1 shows no direct interaction with YTHDC1 but is able to co-precipitate SRSF3 (*Figure 5B*). We further hypothesized that the C-terminus of YTHDC1 interacts with hypo-phosphorylated SRSF3. Indeed, we observed an RNA-independent interaction between endogenous SRSF3 and the C-terminus of YTHDC1, but did not observe an interaction between endogenous SRSF3 and the N-terminus of YTHDC1 (*Figure 5—figure supplement 1B*). Confirmation of these YTHDC1–SRSF3 and SRSF3–NXF1 protein complexes provides a mechanistic basis for incorporation of m$^6$A-modified mRNAs into the mRNA export pathway.

We next tested whether SRSF3 and NXF1 can enrich for m$^6$A by performing CLIP followed by LC-MS/MS of the bound mRNA. We found that both SRSF3 and NXF1 are able to enrich m$^6$A compared to input, with SRSF3 producing the greatest enrichment (*Figure 5C*). Although the protein could enrich for m$^6$A when immunoprecipitated from cells, the RNA recognition motif of SRSF3 showed no selectivity for m$^6$A *in vitro* (*Figure 5—figure supplement 1C and D*).

## SRSF3 is a key adaptor in the export of methylated mRNAs

We further investigated the role of SRSF3 in nuclear export by performing LC-MS/MS on nuclear and cytoplasmic mRNA following knockdown of the protein in HeLa cells. We find that knockdown of SRSF3 leads to a large increase in nuclear m$^6$A levels, whereas other members of the SR protein family that interact with YTHDC1 show only modest changes in m$^6$A levels in the nucleus or cytoplasm upon knockdown (*Figure 5D*, *Figure 5—figure supplement 2A–C*). Depletion of the other

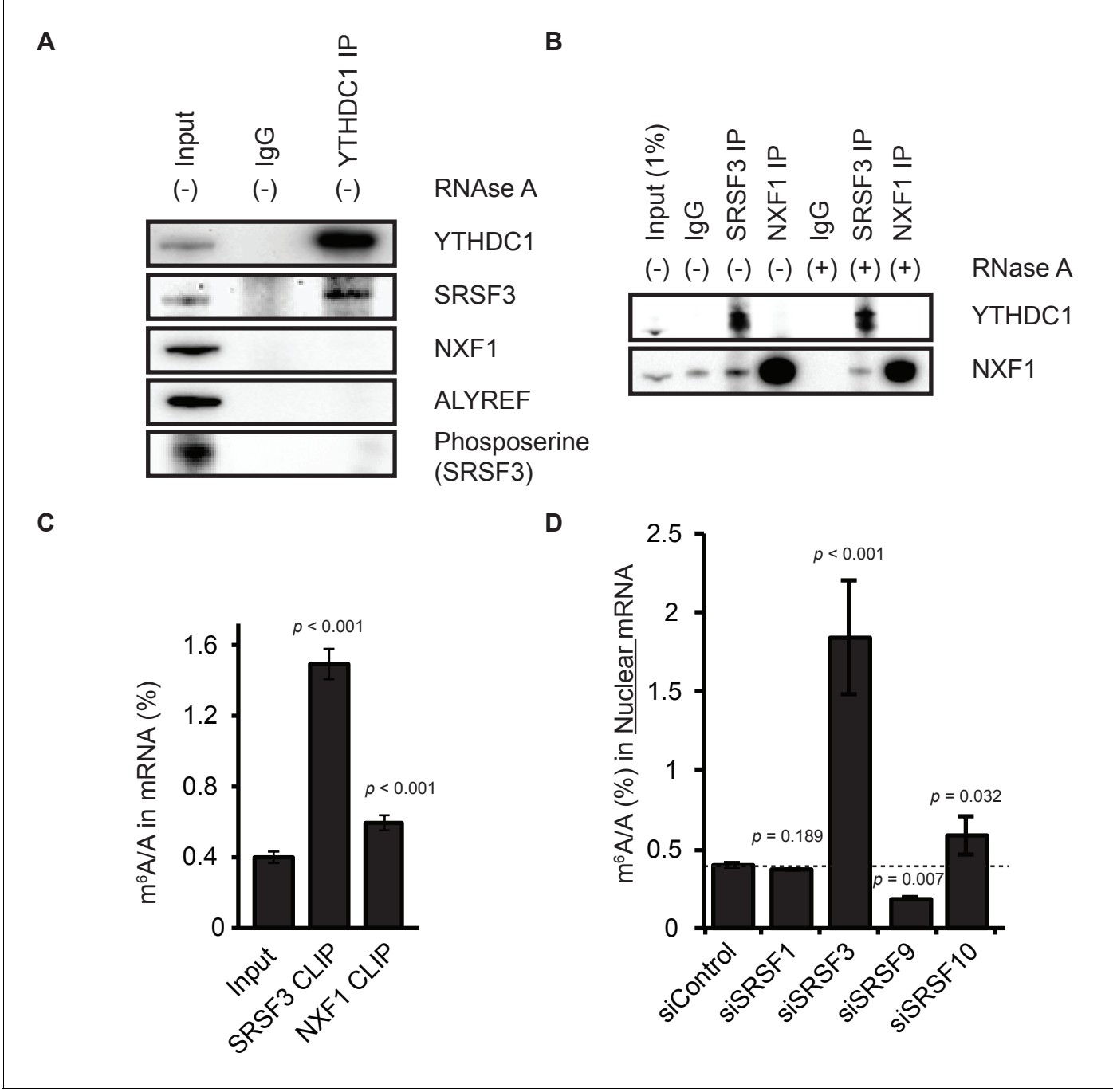

**Figure 5.** YTHDC1 mediates export of m6A-modified mRNA through SRSF3 and NXF1. (**A**) Immunoprecipitation (IP) of endogenous YTHDC1 from HeLa cell lysate. (**B**) IP of endogenous SRSF3 and NXF1 from HeLa cell lysate. (**C**) LC-MS/MS of mRNA cross-linked to SRSF3 and NXF1. Error bars represent mean ± standard deviation, n = 4, two-sided *t*-test with equal variance. (**D**) Quantification of m6A in nuclear mRNA following knockdown of SR proteins. Error bars represent mean ± standard deviation, n = 4, two-sided *t*-test with equal variance.
DOI: https://doi.org/10.7554/eLife.31311.019

The following source data and figure supplements are available for figure 5:

**Source data 1.** YTHDC1 mediates export of m6A-modified mRNA through SRSF3 and NXF1.
DOI: https://doi.org/10.7554/eLife.31311.025

**Figure supplement 1.** YTHDC1 interacts with export-competent SRSF3.
DOI: https://doi.org/10.7554/eLife.31311.020

**Figure supplement 2.** Roles of SRSF3 and other SR proteins in the export of m6A-methylated mRNAs.
*Figure 5 continued on next page*

*Figure 5 continued*

DOI: https://doi.org/10.7554/eLife.31311.021

**Figure supplement 3.** Estimate of relative protein stoichiometry.

DOI: https://doi.org/10.7554/eLife.31311.022

**Figure supplement 4.** Subcellular distribution of YTHDC1 target mRNAs as a function of YTHDC1, SRSF3, and m⁶A.

DOI: https://doi.org/10.7554/eLife.31311.023

**Figure supplement 4—source data 1.** Subcellular distribution of YTHDC1 target mRNAs as a function of YTHDC1, SRSF3, and m6A.

DOI: https://doi.org/10.7554/eLife.31311.024

nuclear reader proteins HNRNPA2B1 and HNRNPC does not significantly affect subcellular m$^6$A levels in a manner consistent with a role in mRNA export (*Figure 5—figure supplement 2A–C*). Like knockdown of YTHDC1, knockdown of SRSF3 results in increased nuclear staining of polyA RNAs and decreased cytoplasmic signal (*Figure 5—figure supplement 2D*). Analyzing clearance of m$^6$A in nuclear mRNA upon knockdown of SRSF3 shows that while initial levels are high, the clearance rate resulting from such a perturbation is rapid, and may rely on mechanisms that have yet to be revealed (*Figure 5—figure supplement 2E*). Taken together, these data suggest that SRSF3 facilitates the export of m$^6$A-methylated mRNAs via interaction with both the m$^6$A-selective protein and YTHDC1 and the canonical mRNA export receptor NXF1, and that SRSF3 may contribute to additional fates for modified nuclear transcripts. According to the results of both RNA-sequencing (our data) and proteomic studies (*Nagaraj et al., 2011*), YTHDC1 is a limiting protein in this system (*Figure 5—figure supplement 3A and B*). Knockdown of YTHDC1, SRSF3, or the methyltransferase components METTL3/14 results in nuclear accumulation of YTHDC1 targets without affecting non-target HPRT1 mRNA (*Figure 5—figure supplement 4A and B*). In the cytoplasm, knockdown of either YTHDC1 or SRSF3 reduces levels of YTHDC1 targets, whereas depletion of the methyltransferase results in significant accumulations – probably as a result of reduced cytoplasmic clearance of transcripts bearing reduced m$^6$A through YTHDF2 (*Figure 5—figure supplement 4C*).

## YTHDC1 facilitates RNA binding to SRSF3 and NXF1

The YTH domain of YTHDC1 has been extensively biochemically characterized as a nuclear 'reader' protein, and binds mRNA with preference for m$^6$A (*Xu et al., 2014*). Thus, we hypothesized that this domain serves as a source of selectivity for mRNA export by facilitating the association of methylated mRNAs with components of the export machinery. Previous work has suggested that YTHDC1 is required for RNA binding by SRSF3, consistent with a role for YTHDC1 in facilitating mRNA export selectivity (*Xiao et al., 2016*). In order to test this functional role in mRNA export, we performed CLIP of SRSF3 followed by LC-MS/MS of bound mRNAs under control and YTHDC1-knockdown conditions. We observed enrichment of m$^6$A in mRNA bound by SRSF3 under both conditions. However, knockdown of YTHDC1 reduced enrichment of m$^6$A by SRSF3, suggesting that YTHDC1 partially facilitates substrate selection by SRSF3 (*Figure 6A*). Sequencing of SRSF3-bound RNAs under control and knockdown conditions further supports this role of YTHDC1, as knockdown of YTHDC1 reduced target transcript enrichment of SRSF3 (*Figure 6B*). Similarly, knockdown of YTHDC1 abolished m$^6$A enrichment by NXF1 as determined by LC-MS/MS (*Figure 6C*) and reduced transcript enrichment as determined by NXF1 RIP-seq (*Figure 6D*). These data show that YTHDC1 is required for proper incorporation of methylated mRNA targets into mRNPs consisting of SRSF3 and NXF1, which together function to facilitate nuclear export of their RNA cargo.

## SRSF3 binding is required for export of YTHDC1 targets

Our data suggest that SRSF3 serves as a key adaptor in mediating the export of mRNA bound by YTHDC1, as SRSF3 and other members of the SR protein family interact directly with the export receptor NXF1 (*Huang et al., 2003*; *Müller-McNicoll et al., 2016*). A prediction of this model is that YTHDC1's targets require association with SRSF3 in order to experience selective export upon binding by YTHDC1. In order to test this, we utilized our RIP-seq data on SRSF3 to analyze the fate of transcripts that are bound by both YTHDC1 and SRSF3 (Group IV), and to compare these transcripts to those that are bound by YTHDC1 but not SRSF3 (Group II) (*Figure 7—figure supplement 1A*). We analyzed subcellular RNA-seq data following knockdown of YTHDC1 according to these

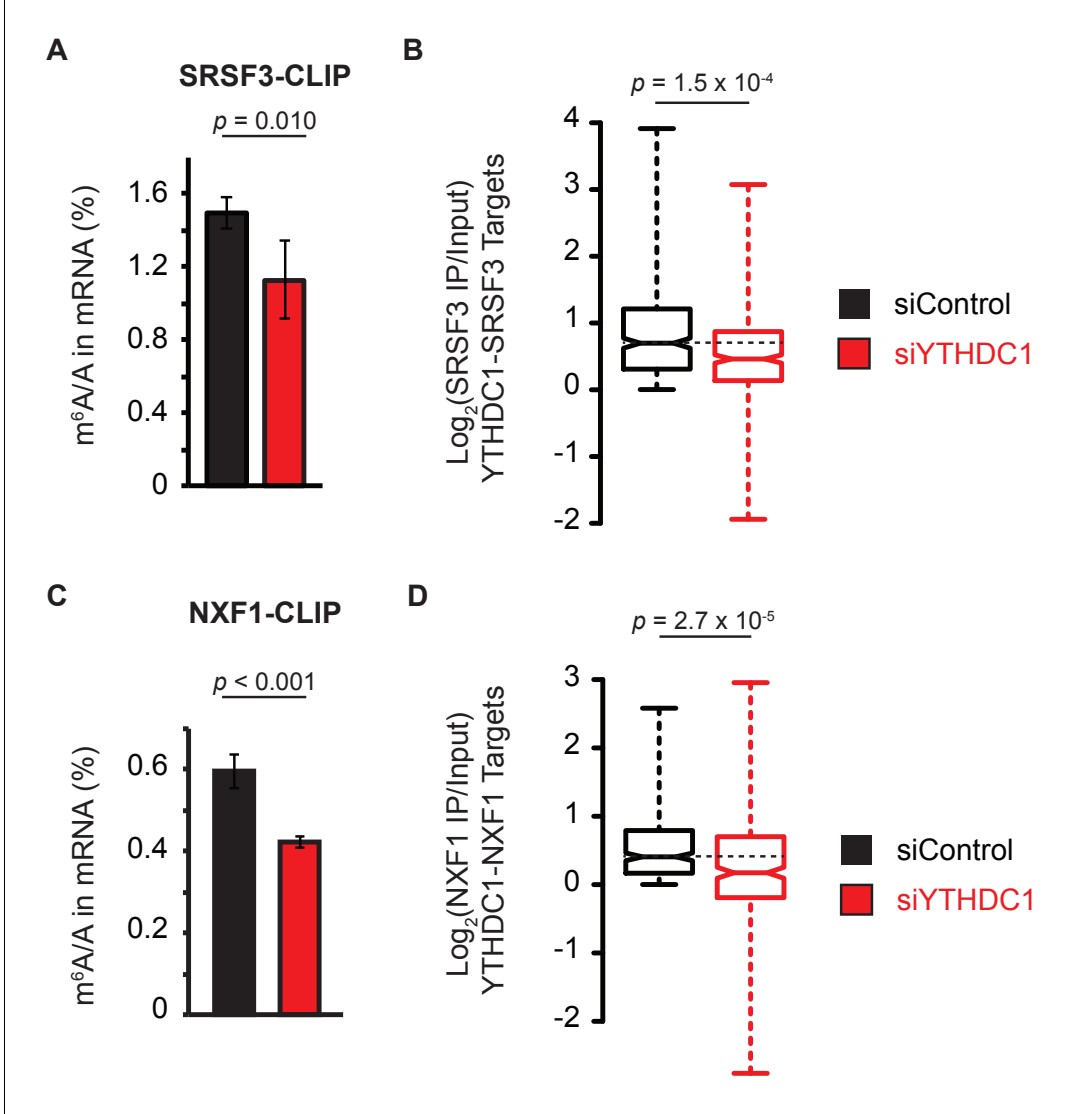

**Figure 6.** YTHDC1 facilitates RNA binding by mRNA export machinery. (**A**) LC-MS/MS of mRNA cross-linked to SRSF3. Error bars represent mean ± standard deviation, n = 4, two-sided *t*-test with equal variance. (**B**) Target enrichment by SRSF3. Whiskers represent three times the interquartile range. Data were analyzed across two independent experiments representing biological replicates. *P*-value calculated using Welch's *T*-test. Whiskers represent three times the interquartile range. (**C**) LC-MS/MS of mRNA cross-linked to NXF1. Error bars represent mean ± standard deviation, n = 4, two-sided *t*-test with equal variance. (**D**) Target enrichment by NXF1. Whiskers represent three times the interquartile range. Data were analyzed across two independent experiments representing biological replicates. *P*-value calculated using Welch's *T*-test. Whiskers represent three times the interquartile range.

DOI: https://doi.org/10.7554/eLife.31311.026

The following source data is available for figure 6:

**Source data 1.** YTHDC1 facilitates RNA binding by mRNA export machinery.

DOI: https://doi.org/10.7554/eLife.31311.027

transcript groupings, and found that only transcripts bound by both YTHDC1 and SRSF3 show significant nuclear accumulation upon YTHDC1 knockdown, whereas transcripts bound by YTHDC1 but not SRSF3 show no significant nuclear accumulation when comparing mean abundance (*Figure 7A*). We observe a similar dependence on SRSF3 binding in cytoplasmic depletion following knockdown of YTHDC1, indicating a functional role for binding by both YTHDC1 and SRSF3 as opposed to YTHDC1 binding alone (*Figure 7—figure supplement 1B*). SRSF3 is a protein that is involved in multiple distinct nuclear functions. Our data suggest that SRSF3 serves as an export adapter for

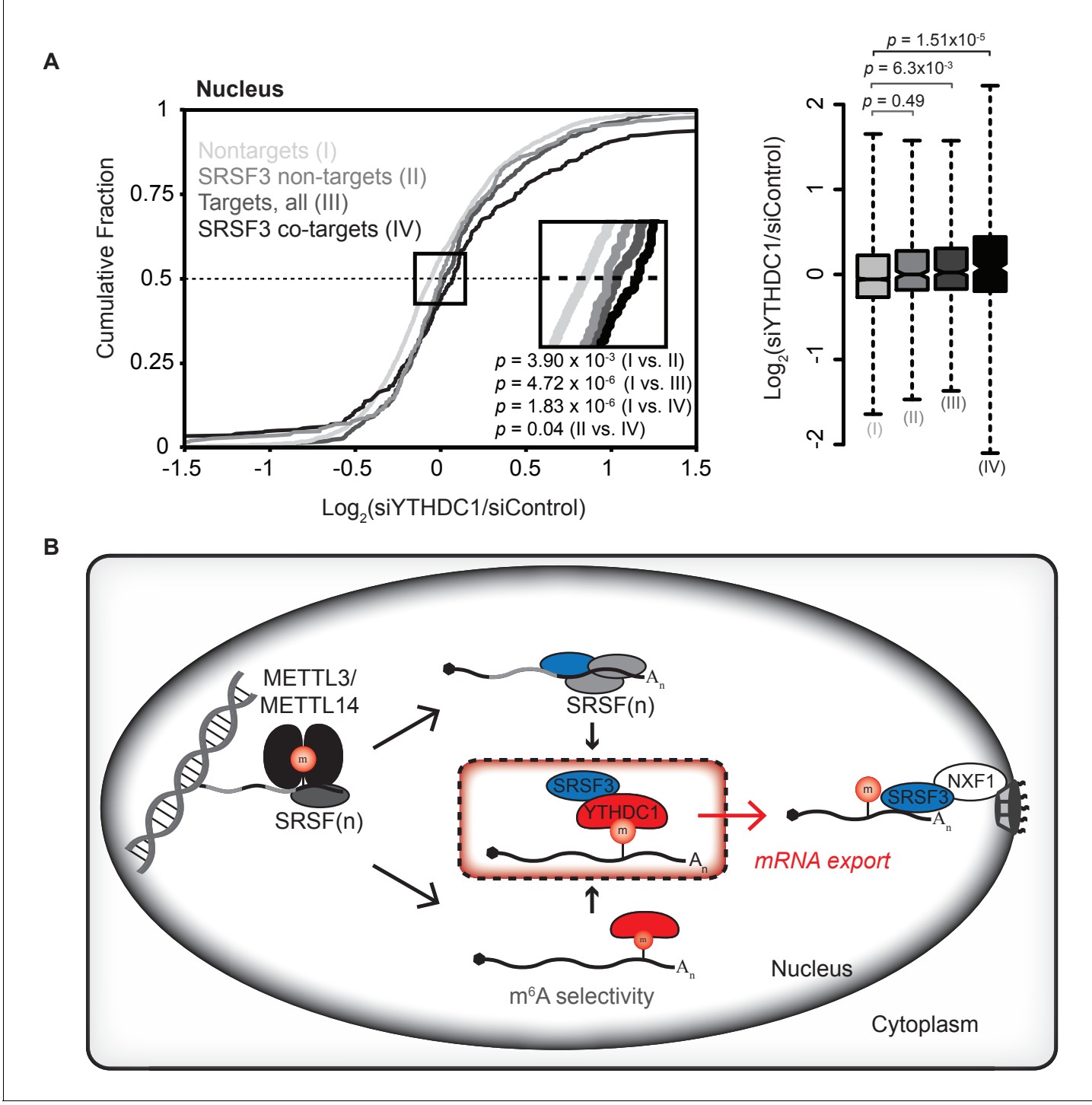

**Figure 7.** Association with SRSF3 is required for facilitated export by YTHDC1. (**A**) Nuclear abundance of transcripts based on binding by YTHDC1 and/ or SRSF3. Cumulative fractions: Mann-Whitney-Wilcoxon Test. Box plot: Welch's *T*-Test. Whiskers represent three times the interquartile range. (**B**) Model of selective export of methylated mRNAs by YTHDC1.

DOI: https://doi.org/10.7554/eLife.31311.028

The following source data and figure supplement are available for figure 7:

**Source data 1.** Association with SRSF3 is required for facilitated export by YTHDC1.
DOI: https://doi.org/10.7554/eLife.31311.030

**Figure supplement 1.** Definition of transcript groups based on YTHDC1 and SRSF3 binding, and effect on cytoplasmic abundance.

*Figure 7 continued on next page*

*Figure 7 continued*

DOI: https://doi.org/10.7554/eLife.31311.029

methylated nuclear mRNAs bound by both YTHDC1 and SRSF3. This study provides a basis for distinct nuclear fates for mRNAs that are bound by YTHDC1 in which a downstream adaptor protein — here, SRSF3 – forms the functional protein–RNA complex for selective association with NXF1 for export to the cytoplasm (*Figure 7B*).

## Discussion

We present a previously unrecognized function for YTHDC1 in mediating nuclear to cytoplasmic mRNA transport of methylated mRNAs. We further provide a mechanistic basis for this phenomenon, in which the methylated mRNA is recognized by the nuclear protein YTHDC1 and delivered to the nuclear mRNA export receptor NXF1 via association with SRSF3. The knockdown of YTHDC1 results in defective nuclear export of target mRNAs, and does so in an $m^6A$-dependent, splicing-independent manner. Binding of YTHDC1 to target transcripts is sufficient to facilitate nuclear export, which indirectly leads to increased translation by increasing the cytoplasmic abundance of its targets. The interaction between YTHDC1 and SRSF3 couples $m^6A$ selectivity to nuclear export, in which $m^6A$ binding by YTHDC1 propagates transport via NXF1 involving the adaptor SRSF3, which itself does not impart biochemical selectivity for $m^6A$ *in vitro*. YTHDC1 knockdown results in decreased association of target mRNAs with both SRSF3 and NXF1, indicating that YTHDC1 is partially responsible for substrate selection by components of the mRNA export pathway. Taken together, these data present a pathway for $m^6A$-dependent nuclear export, and provide evidence for the emerging regulatory capacity of the nuclear mRNA export machinery.

### $m^6A$ selectivity in nuclear mRNA processing and export

Although $m^6A$ methylation in mRNA occurs co-transcriptionally, processes that select for methylated mRNAs within the nucleus are still under investigation. Mechanisms that maintain efficient processing and export of nascent transcripts are critical to enable proper gene expression, given that the vast majority of RNA transcribed in the nucleus is degraded prior to nuclear export (*Moore, 2002*). Previous observations have suggested that cytoplasmic appearance of mRNA is dependent on $m^6A$ methylation (*Camper et al., 1984*). Although YTHDC1 has been shown to affect alternative splicing of select mRNAs (*Rafalska et al., 2004*; *Xiao et al., 2016*; *Zhang et al., 2010*), our data suggest that YTHDC1 can affect the transport of mRNAs from the nucleus in addition to roles in pre-mRNA splicing. These results are consistent with the tight biochemical coupling of mRNA splicing and export, and are reminiscent of the dual roles observed for similar adaptor proteins, particularly those of the SR-family which have been well documented (*Huang and Steitz, 2001*; *Huang et al., 2003*; *Müller-McNicoll et al., 2016*; *Strässer and Hurt, 2001*; *Valencia et al., 2008*; *Zhou et al., 2000*). This work reveals a mechanism by which methylated mRNAs are selectively incorporated into known pathways for mRNA export.

We show that upon knockdown of YTHDC1, mature mRNA localization is perturbed independently of a splicing requirement or splicing outcomes. We propose a model in which YTHDC1 binds to mRNA within the nucleus in an $m^6A$-selective manner, and then delivers target mRNA to SRSF3. Upon association with SRSF3, binding of mRNA by YTHDC1 mediates nuclear export. These results raise questions concerning the recruitment of mRNAs to both YTHDC1 and SRSF3 with respect to splicing. SRSF3 associates with spliceosomal components prior to release by dephosphorylation, at which point the mRNP interacts with the export receptor (*Huang et al., 2004*). Our model suggests that YTHDC1 incorporates bound mRNAs into a dephosphorylated SRSF3-containing complex, facilitating their export. It is possible, however, that the recruitment of shared RNA targets occurs as part of a complex consisting of SRSF3 and YTHDC1, or by recruitment of free SRSF3 by YTHDC1 to methylated regions of long exons and 3' UTRs (*Xu et al., 2014*). In each case, we propose that YTHDC1 binds target RNA and delivers it to SRSF3 in a pathway separate from co-transcriptional binding in association with the spliceosome. Given the relative stoichiometry of YTHDC1 and SRSF3, it is unlikely that YTHDC1 acts entirely upstream of SR-proteins, yet our data suggest that YTHDC1 assists in delivering a subset of SRSF3 RNA cargo, namely methylated mRNAs, for export via NXF1.

Critically, alternative mechanisms could operate for the selective processing of m⁶A-methylated transcripts in which interactions between SR-proteins and components of the methyltransferase during transcription may also contribute to diverse outcomes for methylated transcripts within the nucleus (*Schwartz et al., 2014*), as indicated by enriched nuclear m⁶A methylation upon SRSF3 knockdown. YTHDC1 and SRSF3 do not function exclusively in mediating mRNA export, but the incorporation of SRSF3 to form an export-competent mRNP is critical to facilitate mRNA trafficking to the cytoplasm after binding by YTHDC1.

Depletion of YTHDC1 favors exon skipping, potentially resulting from prolonged nuclear lifetime enabling more extensive pre-mRNA processing. Additionally, perturbation of YTHDC1 levels may affect the localization of SR-proteins (*Xiao et al., 2016*) and prevent access to certain pre-mRNAs. These scenarios may lead to complex outcomes of pre-mRNA processing, the nature of which remains an area for future investigations. In this study, we show that m⁶A methylation serves as a mark for selective nuclear processing, and identify a unique role for YTHDC1 in mediating selectivity in this fundamental component of eukaryotic gene expression.

Current data suggest that m⁶A in mRNA serves to facilitate critical steps in mRNA metabolism, enhancing translation initiation and mRNA decay in the cytoplasm (*Du et al., 2016*; *Lin et al., 2016*; *Wang et al., 2014a*, *2014b*, *, 2015*). Our work shows that selective RNA turnover is common to both nuclear and cytoplasmic mRNA, whereby methylation facilitates mRNA export prior to translation initiation and eventual mRNA decay. Findings presented here contribute to pathways for m⁶A-dependent mRNA metabolism by establishing a mechanistic basis for preferential nuclear processing and export of methylated mRNAs by YTHDC1.

## Materials and methods

### Plasmid construction

λ peptide sequence MDAQTRRRERRAEKQAQWKAAN was fused to the N-terminus of YTHDC1-Flag (*Xu et al., 2014*) by subcloning into pcDNA3.0. YTHDC1 C-Terminal and YTHDC1 N-Terminal regions were similarly subcloned into pcDNA3.0 with C-terminal Flag tags.

λ-YTHDC1 (AA 1–727, EcoRI, XhoI, New England Biolabs, Ipswich, MA)

F: CAGCTTGAATTCATGGACGCACAAACACGACGACGTGAGCGTCGCGCTGAGAAACAAGCTCAATGGAA AGCTGCAAACGGTGGTAGCGCGGCTGACAGTC

R: GCATGCCTCGAGTTACTTGTCATCGTCATCCTTGTAATCTCTCCCCCCTCTTCTATATCGACCTCTCC

λ-N-Terminus (AA 1–355, EcoRI, XhoI)

F: CAGCTTGAATTCATGGACGCACAAACACGACGACGTGAGCGTCGCGCTGAGAAACAAGCTCAATGGAA AGCTGCAAACGGTGGTAGCGCGGCTGACAGTC

R: GCATGCCTCGAGTTACTTGTCATCGTCATCCTTGTAATCTCTCCCCCCTGCATCTTGAAG-CACATATTTG AG

λ-C-Terminus (AA 500–727, EcoRI, XhoI)

F: CAGCTTGAATTCATGGACGCACAAACACGACGACGTGAGCGTCGCGCTGAGAAACAAGCTCAATGGA AAGCTGCAAACGGTGGTAGCTTGTATCAGGTCATTCATAAAATGC

R: GCATGCCTCGAGTTACTTGTCATCGTCATCCTTGTAATCTCTCCCCCCTCTTCTATATCGACCTCTCC

λ-YTHDC1-W377A

Point mutations were introduced using the QuikChange Lightning Site-Directed Mutagenesis Kit (Agilent, Santa Clara, CA) according to the manufacturer's protocol.

F:
GGATCTCCTATACACGCGGTGCTTCCAGCAGGA

R:
TCCTGCTGGAAGCACCGCGTGTATAGGAGATCC

Construction of the λ-control peptide and pmirGlo-Ptight-5BoxB was performed as previously reported (*Wang et al., 2014a*).

His-SRSF (AA1-85) was subcloned from pcDNA3.2 V5-DEST 3XFlag (Addgene Plasmid #46736) (*Auyeung et al., 2013*) into pet28a(+) using the following primers:

F:    ATCGCCATGGATGCATCATCATCATCATCATGGCAGCAGCCATCGTGATTCCTGTCCATTG (NcoI)

R: GCTACTCGAGTTATTTTTCACCATTCGACAGTTCCA (XhoI)

Plasmids with high purity for mammalian cell transfection were prepared using Highspeed Maxiprep or Midiprep Kits (Qiagen, Hilden, Germany).

## Mammalian cell culture, siRNA knockdown and plasmid transfection

The HeLa cell line used in this study was purchased from and validated by ATCC by STR profiling (ATCC CCL-2) and cultured in DMEM (Gibco [ThermoFisher], Waltham, MA, 11965) supplemented with 10% FBS (Gemini Bio Products, West Sacramento, CA) and 1% 100 x Pen Strep (Gibco). Upon revival, the cell lines tested negative for *Mycoplasma* contamination using the LookOut Mycoplasma PCR Detection Kit (Sigma Aldrich MP0035). New cells were thawed periodically every 2–3 months of this study. Transfections were carried out using Lipofectamine 2000, Lipofectamine LTX (for cotransfection) or Lipofectamine RNAiMAX (ThermoFisher) according to the manufacturer's protocol in the absence of antibiotics.

siRNA sequences were purchased from Qiagen (FlexiTube siRNA), listed 5' to 3' where available.
Control: Qiagen All Stars Negative Control (1027281)
YTHDC1 (NM_001031732), Hs_YTHDC1_2
Target sequence: GTCGACCAGAAGATTATGATA
Sense: CGACCAGAAGAUUAUGAUATT
Antisense: UAUCAUAAUCUUCUGGUCGAC
YTHDC1 (NM_001031732), Hs_YT521_3
Target sequence: ATCGAGTATGCAAATATTGAA
Sense: CGAGUAUGCAAAUAUUGAATT
Antisense: UUCAAUAUUUGCAUACUCGAT
SRSF1 (NM_001078166), Hs_SRSF1_5
Target sequence: CTGGACTGCCTCCAAGTGGAA
Sense: GGACUGCCUCCAAGUGGAATT
Antisense: UUCCACUUGGAGGCAGUCCAG
SRSF1 (NM_001078166), Hs_SRSF1_6
Target sequence: TTGGCAGGATTTAAAGGATCA
Sense: GGCAGGAUUUAAAGGAUCATT
Antisense: UGAUCCUUUAAAUCCUGCCAA
SRSF3 (NM_003017), Hs_SRSF3_7
Target sequence: AACCCTAGATCTCGAAATGCA
Sense: CCCUAGAUCUCGAAAUGCATT
Antisense: UGCAUUUCGAGAUCUAGGGTT
SRSF3 (NM_003017), Hs_SRSF3_10
Target sequence: CTCGTAGTCGATCTAGGTCAA
Sense: CGUAGUCGAUCUAGGUCAATT
Antisense: UUGACCUAGAUCGACUACGAG
SRSF9 (NM_003769), Hs_SRSF9_5
Target sequence: AATGGTTATGATTATGGCCAG
Sense: UGGUUAUGAUUAUGGCCAGTT
Antisense: CUGGCCAUAAUCAUAACCATT
SRSF9 (NM_003769), Hs_SRSF9_6
Target sequence: CAGGGCCATATTAGCAGTGAA
Sense: GGGCCAUAUUAGCAGUGAATT
Antisense: UUCACUGCUAAUAUGGCCCTG
SRSF10, Hs_FUSIP1_2
Target sequence: CGGCGTGAATTTGGTCGTTAT
SRSF10, Hs_FUSIP1_7
Target sequence: ATAGAAGATCGTATAGTCCTA
NXF1 (NM_001081491), Hs_NXF1_4
Target sequence: CGCGAACGATTTCCCAAGTTA
Sense: CGAACGAUUUCCCAAGUUATT

Antisense: UAACUUGGGAAAUCGUUCGCG
NXF1 (NM_001081491), Hs_NXF1_5
Target sequence: ACCGAAGGATATCTATCATCA
Sense: CGAAGGAUAUCUAUCAUCATT
Antisense: UGAUGAUAGAUAUCCUUCGGT
YTHDF1, Hs_YTHDF1_1
Target sequence: CCGCGTCTAGTTGTTCATGAA
YTHDF1, Hs_YTHDF1_8
Target sequence: GAGGCTGGAGAATAACGACAA
YTHDF2, Hs_YTHDF2_3
Target sequence: AAGGACGTTCCCAATAGCCAA
HNRNPC, Hs_HNRNPC_1
Target sequence: AACGTCAGCGTGTATCAGGAA
HNRNPC, Hs_HNRNPC_17
Target sequence: CTGATGTGAGCTCATGTTACA
HNRNPA2B1, Hs_HNRNPA2B1_9
Functionally verified against human HNRNPA2B1
ZCCHC8, Hs_ZCCHC8_6
ZCCHC8, Hs_ZCCHC8_7
MTR4, Hs_SKIV2L2_4
Target sequence: ATGACTGGTGATGTTACTATT
MRT4, Hs_SLIV2L2_3
Target sequence: CCAGGTCGTTTGGTAAAGGTA

## Actinomycin D treatment

Actinomycin D (Sigma) was dissolved in DMSO to a final concentration of 5 mg/mL. This stock was diluted 1:1000 (5 µg/mL final concentration) in fresh media and applied to control and YTHDC1 knockdown HeLa cells in culture. Cells were collected by cell lifter at designated time points.

## Cellular fractionation and RNA isolation

HeLa cells were fractionated using NE-PER Nuclear and Cytoplasmic Extraction Reagents (Thermo-Fisher) according to the manufacturer's protocol with the following modifications: following cytoplasmic isolation, nuclei were washed extensively ($4 \times 200$ µL) with PBS. RNA from each portion was collected using Directzol RNA miniprep (Zymo Research, Irvine, CA) and treated with DNaseI prior to elution in water.

RNA was further purified for RNA-sequencing by RiboMinus Eukaryotic Kit v2 (ThermoFIsher) unless otherwise noted, and concentrated using RNA Clean and Concentrator (Zymo Research) for RNA $\geq$200 nt. Sequencing libraries were prepared using the TruSeq Stranded mRNA LT Library Construction Kit (Illumina, San Diego, CA) according to the manufacturer's protocol and sequenced with 50 bp single end reads.

## Western blotting protocol

Protein electrophoresis was performed with NuPAGE Novex 4–12% Bis-Tris Protein Gels (Novex) in MOPS SDS running buffer. Blots were transferred to a nitrocellulose membrane and blocked with 5% milk in TBST (0.05% Tween-20) for 30 min at room temperature. Primary antibodies were incubated overnight in 5% milk in TBST at 4°C over night. Secondary antibodies were incubated at room temperature for one hour, washed and developed using chemiluminescence with SuperSignal West Pico Luminol/Enhancer solution (ThermoFisher) in the FluorChem M system (ProteinSimple, San Jose, CA).

For blots probing for phosphorylation, membranes were blocked using 5% Bovine Serum Albumin (BSA) (EMD Millipore, Billerica, MA), and primary and secondary dilutions were prepared using 5% BSA in TBST.

## Antibodies (target, source, use, dilution, supplier)

Primary

FLAG: mouse monoclonal [M2] – HRP conjugated, western blot, 1:10,000, Sigma Aldrich, St. Louis, MO (A8592)

FLAG: rat polyclonal [L5], IF, 1:300, BioLegend, San Diego, CA (637304)

GAPDH: Goat polyclonal – HRP conjugated, western blot, 1:10,000, GenScript, CHN (A00192-100)

H3: Rabbit polyclonal, western blot, 1:3,000, Abcam, Cambridge, MA (ab1791)

Streptavidin – HRP conjugated, dot blot, 1:10,000, BIO-RAD, Hercules, CA (STAR5B)

YTHDC1: Rabbit polyclonal, western blot, IP, 1:1,000, Abcam, MA (ab122340)

SRSF1: Rabbit monoclonal EPR8239, western blot, 1:10,000, Abcam, MA (ab129108)

SRSF3: Rabbit polyclonal, western blot, 1:1,000, Abcam, MA (ab73891)

SRSF3 (SRp20): Rabbit polyclonal, IP, MBL, MA (Code No. RN080PW)

SRSF9: Rabbit polyclonal, western blot, 1:1,000, Abcam, MA (ab155484)

SRSF10 (FUSIP1): Mouse monoclonal, western blot, 1:1000, Abcam, MA (ab77209)

NXF1: Rabbit monoclonal [EPR8009], western blot, IP 1:5,000, Abcam, MA (ab129160)

YTHDF1: Rabbit polyclonal, western blot, 1:1,000, Abcam, MA (ab99080)

HNRNPC: Rabbit monoclonal [EPNCIR152], western blot, 1:10,000, Abcam, MA (ab133607)

HNRNPA2B1: Rabbit polyclonal, western blot, 1:1,000, Abcam, MA (ab31645)

Digoxigenin: Mouse monoclonal [21H8], IF, 1:300, Abcam, MA (ab420)

ALYREF: Rabbit monoclonal [EPR17942], western blot, 1:5,000, Abcam, MA (ab202894)

ZCCHC8: Rabbit monoclonal [EPR13612], IP, western blot 1:1,000, Abcam, MA (ab181152)

MTR4: Rabbit polyclonal, IP, Abcam, MA (ab70552)

MTR4: Rabbit polyclonal, western blot, 1:1,000, Abcam, MA (ab93337)

Phosphoserine: Rabbit polyclonal, western blot, 1:100, Abcam, MA (ab9332)

### Secondary

Rabbit IgG: Goat polyclonal, western blot, 1:5,000, Bethyl, Montgomery, TX (A120-101P)

Mouse IgG: Goat polyclonal, western blot, 1:5,000, Bethyl, TX (A90-116P)

Mouse IgG: Goat polyclonal – TexasRed, IF, 1:300, Molecular Probes, Eugene, OR (T-6390)

Rat IgG: Donkey polyclonal – Alexa488, IF, 1:300, Molecular Probes, OR (A-21208)

### Northern blotting protocol

RNA from nuclear and cytoplasmic extracts was isolated using Trizol according to the manufacturer's protocol, then further purified using RNA Clean and Concentrator with DNase treatment (Zymo Research). 300 ng of total RNA was diluted 1:2 in 2X TBE-Urea Sample Buffer (ThermoFisher), heated to 65°C for 10 min, and separated on a 6% TBE-Urea gel (180 V, 1 hr) run at 4°C. RNA was transferred at 4°C to an Amersham HyBond-N +membrane in 0.5X TBE Buffer. The membrane was crosslinked using the Stratalinker 2400 (Stratagene, San Diego, CA) autocrosslink option. The membrane was blocked in pre-hybridization buffer (10X final concentration Denhardt's Solution (ThermoFisher)), 6X final concentration SSC Buffer (ThermoFisher), 0.1% SDS, 10 μg/mL salmon sperm DNA (ThermoFisher) at 42°C for 2 hr.

DNA probes (10 μM stock, 2 uL probe per reaction) were labeled with $P^{32}$ using T4 PNK (ThermoFisher) and purified using Oligo Clean and Concentrator (Zymo Research). Labeled oligos were incubated with membrane at 42°C while rotating overnight. Membranes were washed 4 × 1 hr with pre-hybridization buffer at 42°C, then dried at 80°C for 30 min before radioisotope exposure.

Northern probe sequences: tRNAS initiator methionine: TGGTAGCAGAGGATGGTTTCGATCCATCGACCTCTGGGTTATGGGCCCAGCACGCTTCCGCTGCGCCACTCTGCT

U2 snRNA: GAACAGATACTACACTTGATCTTAGCCAA

### Dot blot protocol

HeLa cells were cultured as previously described. Prior to harvesting, 4-thio-uridine was added to the media to a final concentration of 100 μM for 1 hr. Total RNA from Hela cells (Control [−4SU], Control [+4 SU], *YTHDC1*-overexpression, siYTHDC1) was isolated as previously described). RNA was diluted to 250 ng/μL, serially diluted, and spotted on Amersham HyBond-N +membrane. The membrane was crosslinked using the Stratalinker 2400 autocrosslink option. The membrane was

blocked overnight at 4°C in 10% BSA in TBST. After blocking, Streptavidin-HRP (BIO-RAD) was added 1:10,000 and incubated at room temperature for 1 hr. The membrane was washed 8 times for 10 min at room temperature, and exposed using chemiluminescence with SuperSignal West Pico Luminol/Enhancer solution (ThermoFisher) in the FluorChem M system (ProteinSimple, San Jose, CA). Following exposure, the membrane was incubated in methylene blue (Sigma Aldrich) at room temperature for 10 min, and washed briefly with water before imaging.

## RT-PCR

RT-PCR was performed using SuperScript III Platinum One-Step qRT-PCR Kit w/ROX (ThermoFisher) from 50 ng total RNA on an Applied Biosystems 7300 Real-Time PCR System.

GAPDH
F: AGAAGGCTGGGGGCTCATTTG
R: AGGGGCCATCCACAGTCTTC
HPRT1
F: TGACACTGGCAAAACAATGCA
R: GGTCCTTTTCACCAGCAAGCT
Firefly Luciferase
F: CACCTTCGTGACTTCCCATT
R: TGACTGAATCGGACACAAGC
Renilla Luciferase
F: GTAACGCTGCCTCCAGCTAC
R: CCAAGCGGTGAGGTACTTGT

## LC-MS/MS

$2 \times 10^7$ HeLa cells ($2 \times 15$ cm plates) were subjected to 72 hr knockdown using Lipofectamine RNAi-MAX according to the manufacturer's protocol. After 72 hours, cells were washed with PBS and collected with a Cell Lifter (Corning). 10% was used for analysis of total RNA, and the remainder was separated into cytoplasmic and nuclear components. For all samples of mRNA, mRNA was isolated by polyA selection using Dynabeads mRNA DIRECT (ThermoFisher) followed by ribosomal RNA depletion by RiboMinus Eukaryotic Kit v2 (ThermoFisher), then concentrated using RNA Clean and Concentrator (Zymo Research). 100 ng of mRNA was digested using Nuclease P1 (Wako Chemicals, Richmond, VA) at 37°C for 2 hr in a reaction volume of 24 µL. *Escherichia coli* alkaline phosphatase (Sigma Aldrich) was added to the solution along with 5 µL 1M ammonium bicarbonate (aq.) and incubated at 37°C over night. Solutions were diluted 1:2 with water, filtered (0.22 µm) and subjected to LC-MS/MS on an Agilent 6460 Triple Quad MS-MS mass spectrometer coupled to a 1290 UHPLC for multiple reaction monitoring. Nucleotides were separated using a ZORBAX XDB-C18 column (Agilent, 927700–902) with mobile phase of water +0.1% formic acid (Sigma Aldrich) (Buffer A) and methanol +0.1% formic acid. Calibration curves for LC-MS/MS runs were generated prior to each experiment, and used to calculate analyte concentrations for experimental samples. Injections were conducted induplicate and averaged to generate one data point.

## Analysis of high-throughput sequencing data

YTHDC1 PAR-CLIP was performed as previously described (*Xu et al., 2014*). YTHDC1-RIP was performed according to the procedure described in the literature (*Peritz et al., 2006*) using polyA selected RNA and ribosomal-RNA-depleted RNA as biological replicates. mRNA for subcellular RNA-seq was isolated by ribosomal depletion. A summary of sequencing data is provided as *Supplementary file 1*.

Treatment of raw data: reads were trimmed using Cutadapt v.1.4.1 (*Martin, 2011*) and aligned to hg19 using TopHat v.2.0.11 (*Kim et al., 2013*). PAR-CLIP peaks were called using PARalyzer v.1.5 (*Corcoran et al., 2011*) with default parameters. Binding motifs were determined by HOMER v.4.7 (*Heinz et al., 2010*). Differential expression between replicate data sets was calculated using Cufflinks v.2.2.1 (*Trapnell et al., 2012*). Overlapping genomic elements were assessed using BedTools v.2.2.1 (*Quinlan and Hall, 2010*) and analyzed using PeakAnnotator v.1.4 (*Salmon-Divon et al., 2010*). Differential splicing was analyzed using MISO v.0.5.3 (*Katz et al., 2010*).

## Example MISO code

```
#Make indexed gff file after downloading their annotation for hg19
#running for SE (in siYTHDC1_PE150 folder)
$:nohup  python/media/Database/Tools/misopy-0.5.3/misopy/miso.py  -run  hg19/
indexed_SE_events/siYTHDC1_3_PE150_th.out/siYTHDC1-3_PE150_accepted_hits.bam
-output-dir miso/siYTHDC1_3_miso_SE/ -read-len 150 &
#summarize (in miso folder)
$:nohup  python/media/Database/Tools/misopy-0.5.3/misopy/summarize_miso.py  -
summarize-samples siCtl1_miso_SE/siCtl1_miso_SE/siCtl1_SE_miso_summary &
#compare samples (in miso folder)

$:nohup python/media/Database/Tools/misopy-0.5.3/misopy/compare_miso.py -com-
pare-samples  siCtl1_miso_SE/siYTHDC1_2_miso_SE/miso_comparisons/siCtl1_vs_-
siYTHDC1_2_SE &
# For example, to filter the file control.miso_bf to contain only events with: (a)
at least 1 inclusion read, (b) 1 exclusion read, such that (c) the sum of inclusion
and exclusion reads is at least 10, and (d) the ΔΨ is at least 0.20 and (e) the Bayes
factor is at least 10, and (a)-(e) are true in one of the samples, use the following
command:
$:  nohup  python/media/Database/Tools/misopy-0.5.3/misopy/filter_events.py  -
filter         miso_comparisons/siCtl1_vs_siYTHDC1_2_SE/siCtl1_miso_SE_vs_-
siYTHDC1_2_miso_SE/bayes-factors/siCtl1_miso_SE_vs_siYTHDC1_2_miso_SE.
miso_bf -num-inc 5 num-exc 5 -num-sum-inc-exc 10 -delta-psi 0.20 -bayes-factor 10
-output-dir miso_comparisons/siCtl1_vs_siYTHDC1_2_SE_filtered &
# 5 events required for inclusion and exclusion, totaling 10. Psi difference of
0.20 and Bayes factor of at least 10.
#running sashimi_plot
$:python      sashimi_plot.py    -plot-event      "chr5:112157593:112157688:
+@chr5:112159004:112159057:+@chr5:112162805:112162944:+"./sashimi_plot_set-
tings.txt -output-dir./test_plot_1
```

Gene ontology (GO) analysis was conducted using DAVID Functional Annotation Tools v.6.7 (*Huang et al., 2009*).

Statistical comparisons were calculated using the R Statistics package. N values represent biological replicates unless otherwise stated.

Raw and processed data files have been deposited in the Gene Expression Omnibus (GEO, http://www.ncbi.nlm.nih.gov/geo) and are accessible under GSE74397.

A summary of sequencing experiments can be found in *Supplementary file 1*.

### Immunofluorescence

HeLa cells were seeded in eight chamber microscope slides (Nunc 155409) and treated with siRNA as previously described. After 72 hr, cells were washed with PBS, then fixed with 4% paraformaldehyde (Sigma) in PBST (PBT +0.05% Tween-20), freshly prepared by heating at 65° C until clear, at RT for 15 min. Fixative was removed and chilled (−20°C) methanol was added dropwise and incubated at RT for 15 min. Cells were washed with PBS and blocked with 10% FBS in PBST at RT for 1 hr. Primary antibody incubation was performed in 10% FBS in PBST at 4°C overnight. After 4 × 5 min washes with PBST secondary antibody incubation was performed at RT for 1 hr. After 4 × 5 min washes with PBST, cells were treated with SlowFade Gold antifade reagent with DAPI (Molecular Probes) and imaged using a Leica TCS SP5 II STED laser scanning confocal microscope (Leica Microsystems, Inc.).

## PolyA imaging

HeLa cells were fixed in PFA and MeOH as above. Following methanol treatment, cells were washed with PBST. PBST was removed and RISH-positive control DIG probe (PanPath Q152P.9900) (37-mer oligonucleotide complementary to PolyA) was added as a 1:3 dilution in PBST and incubated at 37° C for 2 hr. Cells were washed four times for 5 min with PBST at RT and subjected to an IF staining protocol. Image quantification was performed using Fiji (*Schindelin et al., 2012*).

## Nascent RNA-labeling

Nascent RNA labeling was conducted using the Click-iT Nascent RNA Capture Kit (Molecular Probes) according to the manufacturer's protocol. Briefly, HeLa cells treated with siRNA were fed 5-ethynyl-urudine (EU) at a concentration of 200 µM for 4 hr. After 4 hr, RNA wasisolated from whole cells (10%) and nuclear and cytoplasmic portions. Click chemistry was performed on RNA, precipitated in ethanol and enriched using provided streptavidin beads. cDNA synthesis was performed on the beads using SuperScript VILO (Invitrogen). RT-PCR was performed using FastStart Essential DNA Green Master in a LightCycler 96 (Roche).

## RT-PCR primer sequences

APC
 F: TAGGGGGACTACAGGCCATT
 R: TTTAGTTGGGCCACAAGTGC
 MCL1
 F: CGGACTCAACCTCTACTGTGG
 R: CTTGGAAGGCCGTCTCGT
 SOX12
 F: GCTGAGGAAGGTGAAGAGGA
 R: GCGATCATCTCGGTAACCTC
ACTB
 F: ACAGAGCCTCGCCTTTGCC
 R: GATATCATCATCCATGGTGAGCTGG
GAPDH
 F: AGAAGGCTGGGGCTCATTTG
 R: AGGGGCCATCCACAGTCTTC
HPRT1
 F: TGACACTGGCAAAACAATGCA
 R: GGTCCTTTTCACCAGCAAGCT

## Luciferase reporter assay

All reporter experiments were conducted in HeLa Tet-off cells (Clontech, 631156) cultured in DMEM with 1% 100 x Pen Strep (Gibco) supplemented with 10% Tet-System approved FBS (Clontech, 631106). Cultures were maintained in 200 µg/mL G418 (Takara) and tested negative for *Mycoplasma* contamination using the LookOut Mycoplasma PCR Detection Kit (Sigma Aldrich MP0035).

In a six well plate, ~1 × 10$^6$ cells were cultured with 100 ng/mL doxycycline (Dox). Cells were transfected with 50 ng reporter plasmid and 450 ng effector plasmid using Lipofectamine LTX according to manufacturer's protocol. The following day, cells were washed with PBS, trypsinized and washed thoroughly (4 × 15 mL PBS) to induce luciferase expression. Washed pellets were suspended in 3 mL culture media and seeded in 96-well plates. After 2 hr of induction, Dox was added to a final concentration of 500 ng/mL to halt firefly transcription. Luciferase was quantified using the Dual-Glo Luciferase Assay System (Promega, Madison, WI) in a Synergy HTX Multi-Mode Reader (BioTek). Each sample was normalized to Renilla luciferase signal from the same well.

## Isoform analysis by RT-PCR

RNA was collected as previously described, including treatment with DNase. Reverse transcription was conducted with SuperScript VILO according to the manufacturer's protocol, and PCR was conducted using primer sequences designed to span the region in question using Pfusion High Fidelity

PCR Master Mix (NEB). Fragments were separated on 3.0% agarose and visualized using ethidium bromide (Sigma Aldrich).

Primer Sequences:

APC

F: TTCCTTACAAACAGATATGACCAGA

R: TTACCAGAAGTTGCCATGTTG

Product observed: 210 bp

MCL1

F: GAGGAGGACGAGTTGTACCG

R: ACCAGCTCCTACTCCAGCAA

Product observed: 525 bp

## Protein co-immunoprecipitation

HeLa cells transfected with empty plasmid or YTHDC1-Flag were washed with PBS twice, then collected using a cell lifter and spun at 2000 x g for 5 min. Pellets were suspended in two volumes (compared to pellet) Buffer A (10 mM HEPES, pH 7.5, 1.5 mM $MgCl_2$, 10 mM KCl, 0.5 mM DTT), vortexed briefly, and incubated on ice for 15 min. NP-40 was added to a final concentration of 0.25%, vortexed, and left on ice for 5 min. Suspensions were spun at 2000 x g for 3 min at 4°C. The supernatant (cytoplasmic extract) was removed and saved. Nuclei were suspended in two volumes of Buffer B (20 mM HEPES, pH 7.5, 0.42 M KCl, 4 mM $MgCl_2$, 1 mM EDTA, 0.5 mM DTT, 10% glycerol), vortexed briefly, and incubated on ice for 30 min. The suspension was spun at 15,000 x g for 15 min at 4° C, and supernatant combined with the cytoplasmic extract. This was incubated on ice for 15 min and spun again at 15,000 x g for 15 min at 4° C. The supernatant was removed and used as the HeLa lysate.

FLAG IP: lysate from mock and YTHDC1-transfected HeLa cells was incubated with M2-anti-FLAG beads (Sigma) and rotated at 4° C for two hours. After two hours, the beads were washed 4 × 500 uL with Buffer C (wash buffer) (50 mM Tris-HCl, pH 7.5, 100 mM KCl, 5 mM $MgCl_2$, 0.2 mM EDTA, 0.1% NP-40, 10 mM β-ME, 10% glycerol). The beads were resuspended and half were used for western blot analysis. The remaining half was warmed to 37° C and treated with RNase A (Thermo Fisher) at a final concentration of 1 ug/mL for 30 min. The beads were washed with 4 × 500 uL wash buffer and used for western blot analysis.

Endogenous IP: lysate from untreated HeLa cells was treated with rabbit IgG control or primary antibody against SRSF3 or NXF1 (5 ug/IP) and incubated at 4°C for two hours. After two hours, washed Dynabeads Protein A (Thermo Fisher) were added, and the solution was incubated for an additional hour, followed by washing and RNase treatment for western blotting as described.

SRSF3 and NXF1 RIP-seq in the presence and absence of YTHDC1 was conducted following immunoprecipitation under these conditions and subsequent Ribo(-) treatment of isolated RNA.

## CLIP-LC-MS/MS

Crosslinking was performed using the Stratalinker 2400 (Stratagene) autocrosslink option on untreated HeLa cells. Protein co-IP was performed as described, and washed beads were resuspended in 100 uL wash buffer. 100 uL 2X Proteinase K buffer (100 mM Tris-HCl, pH 7.5, 200 mM NaCl, 2 mM EDTA, 1% SDS) was added, followed by Proteinase K (Thermo Fisher) to a final concentration of 500 ug/mL and heated for 1 hr at 65°C. RNA was isolated using TriZol. Recovered RNA was purified using two rounds of poly(A)-selection, digested, and subjected to LC-MS/MS.

## Protein expression and purification from *E. coli*

His-SRSF3-AA1-85, constituting the RRM of SRSF3, was transformed into NEB T7-Express competent *E. coli* (NEB), expanded to 4L and shaken at 37°C. When the optical density reached 0.6–0.8, IPTG was added to a final concentration of 2 mM and the culture was cooled to 16°C overnight. Cultures were collected by centrifugation and resuspended in lysis buffer (20 mM Tris-HCl, pH 7.5, 400 mM NaCl, 1 mM PMSF) for sonication. Sonication was performed on ice at 25% amplitude for 15 min (10 s on, 10 s off, 30 min total). The supernatant was collected after centrifugation at 13,000 x g for 30 min at 4°C and filtered (0.22 um). The supernatant was applied to Ni-NTA resin (GE Healthcare) by gravity flow and washed extensively with wash buffer (lysis buffer + 30 mM imidazole), and eluted in

elution buffer (lysis buffer + 500 mM imidazole). The eluate was concentrated and further purified by Superdex 75 gel filtration in a buffer containing 10 mM Tris-HCl, pH 7.5, 150 mM NaCl and 1 mM DTT. The protein was concentrated to ~2.5 mg/mL and used for *in vitro* IP.

### *In vitro* IP

Purified recombinant His-SRSF3-AA1-85 was diluted to a final concentration of 500 nM in IPP buffer (10 mM Tris-HCl pH 7.5, 150 mM NaCl, 0.1% NP-40, 0.5 mM DTT and 20 U/uL SUPERase-In RNase Inhibitor along with 800 ng HeLa mRNA fragmented with RNA fragmentation reagents [Thermo-FIsher]). Protein and RNA were mixed at 4°C for 2 hr. Dynabeads His-Tag Isolation and Pulldown beads (ThermoFisher, 20 uL) were added after being washed four times and resuspended in 40 uL IPP buffer. The mixture was rotated for an additional two hours. The beads were immobilized and the aqueous phase (flow through) precipitated by ethanol precipitation. The beads were washed four times in 500 uL IPP buffer and homogenized in TriZol reagent. RNA was collected and digested for LC-MS/MS to quantify $m^6A$ in each sample.

### Accession numbers

Raw and processed data files have been deposited in the Gene Expression Omnibus (GEO, http://www.ncbi.nlm.nih.gov/geo) and are accessible under GSE74397. A summary of all sequencing samples can be found in *Supplementary file 1*.

## Acknowledgements

This work was supported by the National Institutes of Health (HG008688, GM113194, both to CH) and the National Natural Science Foundation of China (31171377 and 31471400 to XH). CH is an investigator of the Howard Hughes Medical Institute (HHMI). IAR is supported by the National Institutes of Health Ruth L Kirschstein National Research Service Award (F30GM117646 from NIGMS). The Mass Spectrometry Facility of the University of Chicago is funded by the National Science Foundation (CHE-1048528). Sequencing was performed at the University of Chicago Genomics Facility. Imaging was performed at the University of Chicago Integrated Light Microscopy Facility.

## Additional information

### Funding

| Funder | Grant reference number | Author |
|---|---|---|
| National Institute of General Medical Sciences | F30GM117646 | Ian A Roundtree |
| Howard Hughes Medical Institute | | Ian A Roundtree<br>Guan-Zheng Luo<br>Zijie Zhang<br>Xiao Wang<br>Chuan He |
| National Science Foundation | CHE-1048528 | Ian A Roundtree<br>Guan-Zheng Luo<br>Zijie Zhang<br>Xiao Wang<br>Laura Guerrero<br>Phil Xie<br>Emily He<br>Chuan He |
| National Institute of General Medical Sciences | HG008688 | Chuan He |
| National Institute of General Medical Sciences | GM113194 | Chuan He |
| National Natural Science Foundation of China | 31171377 | Xingxu Huang |
| National Natural Science Foundation of China | 31471400 | Xingxu Huang |

The funders had no role in study design, data collection and interpretation, or the decision to submit the work for publication.

## Author contributions
Ian A Roundtree, Conceptualization, Resources, Data curation, Software, Formal analysis, Supervision, Funding acquisition, Validation, Investigation, Visualization, Methodology, Writing—original draft, Writing—review and editing; Guan-Zheng Luo, Software, Formal analysis, Investigation; Zijie Zhang, Data curation, Investigation; Xiao Wang, Conceptualization, Data curation; Tao Zhou, Yiquang Cui, Jiahao Sha, Xingxu Huang, Bin Shen, Validation, Investigation; Laura Guerrero, Phil Xie, Emily He, Data curation; Chuan He, Conceptualization, Resources, Funding acquisition, Methodology, Project administration, Writing—review and editing

## Author ORCIDs
Chuan He 🔟 http://orcid.org/0000-0003-4319-7424

## Decision letter and Author response
Decision letter https://doi.org/10.7554/eLife.31311.039
Author response https://doi.org/10.7554/eLife.31311.040

# Additional files

## Supplementary files
• Supplementary file 1. Summary of high-throughput sequencing experiments. Related to *Figure 3*, *Figure 6*, *Figure 7*.
DOI: https://doi.org/10.7554/eLife.31311.031
• Transparent reporting form
DOI: https://doi.org/10.7554/eLife.31311.032

## Major datasets
The following dataset was generated:

| Author(s) | Year | Dataset title | Dataset URL | Database, license, and accessibility information |
|---|---|---|---|---|
| Ian A Roundtree, Guan-Zheng Luo, Chuan He | 2015 | N6-methyladenosine-Mediated Nuclear Export of Messenger RNA | https://www.ncbi.nlm.nih.gov/geo/query/acc.cgi?acc=GSE74397 | Publicly available at the NCBI Gene Expression Omnibus (accession no: GSE74397) |

The following previously published datasets were used:

| Author(s) | Year | Dataset title | Dataset URL | Database, license, and accessibility information |
|---|---|---|---|---|
| Wang X, Lu Z, Yue Y, Han D, Hon GC, Ren B, Pan T, He C | 2013 | N6-methyladenosine-Mediated Nuclear Export of Messenger RNA | https://www.ncbi.nlm.nih.gov/geo/query/acc.cgi?acc=GSE49339 | Publicly available at the NCBI Gene Expression Omnibus (accession no: GSE49339) |
| Alarcon CR, Goodarzi H, Lee H, Liu X, Tavazoie S, Tavazoie SF | 2015 | Nuclear HNRNPA2B1 HITS-CLIP and RNA-seq | https://www.ncbi.nlm.nih.gov/geo/query/acc.cgi?acc=GSE70061 | Publicly available at the NCBI Gene Expression Omnibus (accession no: GSE70061) |

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
