## [Decision Letter]

Thank you for submitting your article "N6-methyladenosine and YTHDC1 Promote Nuclear Messenger RNA Export" for consideration by *eLife*. Your article has been reviewed by three peer reviewers, one of whom is a member of our Board of Reviewing Editors and the evaluation has been overseen by Philip Cole as the Senior Editor. The reviewers have opted to remain anonymous.

The reviewers have discussed the reviews with one another and the Reviewing Editor has drafted this decision to help you prepare a revised submission.

This paper describes the role of the m6A reader YTHDC1 in the promotion of mRNA nuclear export. While this mark has already been linked to both pre-mRNA splicing and micro-processing in the nucleus as well as the regulation of translation in the cytoplasm, this study now implicates this mark in mRNA nuclear export. Overall, this study presents interesting and important data. However, it is marred by sub-optimal presentation with a general weakness in explaining the wide range of methodology employed and indeed how the figures can be interpreted. As well as recommending some additional clarifying experiments, we request a major overhaul of the figure description and presentation (in text, legends and Methods).

Our specific suggestions for revision are as follows:

1) The title is inaccurate. The authors describe here an mRNA export pathway, which is not very different from 'canonical' mRNA export. The difference is that methylated mRNA (m6A) is exported to the cytoplasm, with the reader YTHDC1 acting as an adaptor to recruit SRSF3. We don't think that this study clearly demonstrates that methylation of mRNA per se increases the rate of export for an mRNA. This needs clarification, including a more accurate title.

2) Do other m6A readers also affect nuclear export? For instance, in Figure 1, what is the effect of knocking-down other nuclear readers, such as hnRNP A2/B1 or hnRNP C? A potential role for other nuclear readers in this m6A mRNA export pathway should be tested, or at least discussed.

3) What is the relative nuclear stoichiometry of YTHDC1 and SRSF3? Are any of these factors limiting? In other words, what determines that out of all transcripts bound by YTHDC1 only some of them are also bound by SRSF3?

4) The LC-MS/MS method to distinguish A from m6A needs explanation and some of the raw data should be shown in supplemental figures. Really we would like some specific mRNAs to be monitored for nuc/cyt distribution to show their nuclear export dependence on m6A, YTHDC1 and SRSF3

5) The data shown in Figure 5—figure supplement 2 is a bit worrisome. In A, relatively little accumulation of nuclear m6A-containing RNA in YTHDC1 knockdown cells was detected relative to that in SRSF3 knockdown cells. This does not seem to support the model in which YTHDC1 is the first to act on m6A-containing mRNAs and then recruit SRSF3 for their export. Notably SRSF3 has been previously implicated in exporting both mRNAs and histone mRNAs by the Steitz group. Consequently, SRSF3 depletion would be expected to affect mRNA export with or without m6A. These issues need to be addressed.

6) Most mRNAs are known to be either rapidly degraded or exported after transcription in minutes, not hours, based on single molecular analysis from Rob Singer's lab. The results reported here, which are based on either total RNAs or polyA-selected RNAs (not clearly stated in the text), suggest that the average half-life is between 10 and 20 hrs. How can this difference be accounted for?

---

## [Author Response]

This paper describes the role of the m6A reader YTHDC1 in the promotion of mRNA nuclear export. While this mark has already been linked to both pre-mRNA splicing and micro-processing in the nucleus as well as the regulation of translation in the cytoplasm, this study now implicates this mark in mRNA nuclear export. Overall, this study presents interesting and important data. However, it is marred by sub-optimal presentation with a general weakness in explaining the wide range of methodology employed and indeed how the figures can be interpreted. As well as recommending some additional clarifying experiments, we request a major overhaul of the figure description and presentation (in text, legends and Methods).

We thank the reviewers for their thoughtful comments. We have made changes to our manuscript that we feel addresses the weakness in describing methodology and interpreting results. We have also revised our description of the working model to clarify the contribution of this work to the field, and improved our data presentation by emphasizing experimental details in main figures (Nuclear vs Cytoplasmic mRNA) throughout.

Our specific suggestions for revision are as follows:1) The title is inaccurate. The authors describe here an mRNA export pathway, which is not very different from 'canonical' mRNA export. The difference is that methylated mRNA (m6A) is exported to the cytoplasm, with the reader YTHDC1 acting as an adaptor to recruit SRSF3. We don't think that this study clearly demonstrates that methylation of mRNA per se increases the rate of export for an mRNA. This needs clarification, including a more accurate title.

We thank the reviewers for their input. We have changed the title to avoid potential confusion and overstatement. We have also made changes throughout the manuscript to be consistent (changing “promote” to “mediate” nuclear export). We agree that most of our data showed that YTHDC1 preferentially binds methylated mRNA and mediates their nuclear export, which is accomplished via interactions with known export machinery.

We have changed the title to “YTHDC1 Mediates Nuclear Export of N6-methyladenosine Methylated mRNAs**”**.

2) Do other m6A readers also affect nuclear export? For instance, in Figure 1, what is the effect of knocking-down other nuclear readers, such as hnRNP A2/B1 or hnRNP C? A potential role for other nuclear readers in this m6A mRNA export pathway should be tested, or at least discussed.

See Figure 5—figure supplement 2, and subsection “SRSF3 is a key adaptor in export of methylated mRNAs”.

We have studied both HNRNPA2B1 and HNRNPC by LC-MS/MS upon knockdown of these proteins, in addition to several members of the SR-protein family. We also analyzed subcellular RNA sequencing following depletion of HNRNPA2B1. In each case, we do not observe significant changes in subcellular mRNA abundance but maintain that other m6A-binding proteins may influence nuclear trafficking of mRNAs. It is possible that other nuclear reader proteins couple their functions to components of the mRNA export machinery but we have yet to observe effects as noticeable as YTHDC1.

3) What is the relative nuclear stoichiometry of YTHDC1 and SRSF3? Are any of these factors limiting? In other words, what determines that out of all transcripts bound by YTHDC1 only some of them are also bound by SRSF3?

We have addressed this point in subsection “SRSF3 is a key adaptor in export of methylated mRNAs”, as well as the Discussion section.

We can estimate the abundance of YTHDC1, SRSF3, SRSF10, and NFX1 by RNA-sequencing to determine relative stoichiometry. Our sequencing data suggest that YTHDC1 is limiting, as both SRSF3 (3.1x) and SRSF10 (1.7x) mRNA are significantly more abundant. We have also analyzed their relative stoichiometry from proteomic data of HeLa cells (Nagaraj et al., 2011), and find that both SRSF3 and SRSF10 are more abundant that YTHDC1 by at least two orders of magnitude. These data have been added as Figure 5—figure supplement 3. These data are consistent with SRSF3 playing the largest role in mRNA export of the SR-family proteins (Müller-McNicoll et al., 2016). Given its relatively low abundance, YTHDC1 may be limiting in facilitating methyl-selectivity into the mRNA export pathway.

However, these data do not provide insight into which targets of YTHDC1 are bound by SRSF3 and which are not. We have shown that YTHDC1 interacts with SRSF3 lacking serine phosphorylation, and suggest that the stoichiometry of SRSF3 phosphorylation may influence association with YTHDC1.

It is important to note that SRSF3 is assembled into spliceosomes co-transcriptionally in a phosphorylated form, and likely binds its mRNA targets at this time. Dephosphorylation facilitates release form the splicing machinery and allows association with NXF1. YTHDC1 may associate with SRSF3 which lacks phosphorylation as well as an associated RNA target. This population of SRSF3 may be the limiting factor for YTHDC1-mediated incorporation of mRNA into the export pathway. The association of YTHDC1 and the many adaptor proteins with which it is known to bind is certainly a continuing subject of investigation, and may be critical in determining the regulation of its many known functions.

4) The LC-MS/MS method to distinguish A from m6A needs explanation and some of the raw data should be shown in supplemental figures. Really we would like some specific mRNAs to be monitored for nuc/cyt distribution to show their nuclear export dependence on m6A, YTHDC1 and SRSF3

We have included raw data for our LC-MS/MS method as part of Figure 1—figure supplement 1 and E in the form of a representative standard curve and determination of analyte concentration and have expanded our description in subsection “LC–MS/MS”.

For each experiment, a standard curve is generated for each modification of interest, as shown, with each standard run in triplicate. Unknown samples are digested as described in the methods section, and analyzed according to the standard curve. Typically, injections are run in duplicate, and the average of the injections is used as a data point for a given biological replicate. For the unknown shown in the supplementary data, each injection was run in triplicate. The source data for this representative standard curve is provided. In our hands, this technique is much more accurate than other methods used to quantify modification abundance such as dot-blots or 2D-TLC, and is capable of detecting more subtle changes in methylation level.

We have monitored nuclear and cytoplasmic distribution for several YTHDC1 targets to highlight their dependence on YTHDC1, SRSF3, and METTL3/14 (m6A), and discussed these results in subsection “SRSF3 is a key adaptor in export of methylated mRNAs”, and added Figure 5—figure supplement 4. We have included the source data and calculated p-values as a supplementary file of source data.

As expected, we observe significant nuclear accumulation in the absence of these factors, consistent with a role in nuclear clearance of these mRNAs. For both YTHDC1 and SRSF3, this is accompanied by a decrease in cytoplasmic abundance. However, knockdown of the methyltransferase leads to an increase in cytoplasmic transcript levels of these targets. This is most likely a result of m6A-dependent mR.

5) The data shown in Figure 5—figure supplement 2 is a bit worrisome. In A, relatively little accumulation of nuclear m6A-containing RNA in YTHDC1 knockdown cells was detected relative to that in SRSF3 knockdown cells. This does not seem to support the model in which YTHDC1 is the first to act on m6A-containing mRNAs and then recruit SRSF3 for their export. Notably SRSF3 has been previously implicated in exporting both mRNAs and histone mRNAs by the Steitz group. Consequently, SRSF3 depletion would be expected to affect mRNA export with or without m6A. These issues need to be addressed.

We thank the reviewers for their insight. Indeed, the increase in nuclear m6A level upon knockdown of SRSF3 is the largest we have observed in the lab striking. This result suggests to us that SRSF3 plays additional roles in the processing of nuclear mRNAs, likely through direct association with the methyltransferase component METTL14. This is supported by our data in Figure 6, which shows that SRSF3 is able to enrich m6A in mRNA with or without YTHDC1. Taken together, we these data suggest that SRSF3 is capable of binding RNA independent of YTHDC1. However, our data indicate that a subset of SRSF3-bound mRNAs are recruited by YTHDC1.

We have revised our manuscript to reflect this, stating that our data show YTHDC1 facilitates mRNA binding by SRSF3, but may not necessarily act upstream (Discussion section). Given the stoichiometry of SRSF3 compared to YTHDC1, and it’s known interaction with splicing machinery it is unlikely that YTHDC1 acts upstream of the majority of SRSF3 RNA binding.

SRSF3 is known to function as an mRNA export adaptor for its target mRNA, as reported initially by the Steitz group and characterized more recently by Neugebauer lab (Müller-McNicoll et al., 2016). In their report, histone mRNAs show a significant increase in nuclear to cytoplasmic ratio following knockdown of SRSF3 as anticipated. In our studies we focused on and showed data for poly(A)-tailed RNAs. It will be interesting to study the methylation of histone mRNAs in the future. Some of them do get methylated.

6) Most mRNAs are known to be either rapidly degraded or exported after transcription in minutes, not hours, based on single molecular analysis from Rob Singer's lab. The results reported here, which are based on either total RNAs or polyA-selected RNAs (not clearly stated in the text), suggest that the average half-life is between 10 and 20 hrs. How can this difference be accounted for?

We apologize for the confusion in our description of the data. For all LC-MS/MS experiments, we isolate mRNA by polyA selection followed by ribosomal RNA depletion unless otherwise noted.

We apologize for this confusion, and have clarified this point in the Results section as well as the Material and methods section of the manuscript.

We believe the difference in experimental design can explain these apparent discrepancies. In our assay, we treat cells with ActD for 0, 3, and 6 hours before isolating nuclear and cytoplasmic mRNA. We then analyze the remaining mRNA for m6A content, and observe that methylation level decreases over time. These results report a measure of “m6A-halflife”, which is largely independent of the half-life of a given mRNA. We are able to measure m6A/A in mRNA from different starting amounts of total RNA by selecting for polyA transcripts and normalizing the samples to fall within our quantitative range for LC-MS/MS. It is the change in the m6A:A ratio that suggests selective clearance of methylated mRNAs in the nucleus and cytoplasm. Because the ratio of m6A/A is changing more slowly than the clearance of individual mRNAs, we report clearance rates on the order of hours for m6A methylation.

We have clarified this in the text and use the term “apparent half-life of methylation” to describe these results (Results section).